# Higher Dose Oral Fluconazole for the Treatment of AIDS-related Cryptococcal Meningitis (HIFLAC)—*report of A5225, a multicentre, phase I/II, two-stage, dose-finding, safety, tolerability and efficacy randomised, amphotericin B-controlled trial of the AIDS Clinical Trials Group*

Umesh G. Lalloo [1]* , Lauren Komarow[2], Judith A. Aberg[3], David B. Clifford[4],
Evelyn Hogg[5], Ashley McKhann[2], Aggrey Bukuru[6‡], David Lagat[7‡], Sandy Pillay[1‡],
Vidya Mave[8‡], Khuanchai Supparatpinyo[9‡], Wadzanai Samaneka[10‡], Deborah Langat[11‡],
Eduardo Ticona[12‡], Sharlaa Badal-Faesen[13‡], Robert A. Larsen[14‡], the ACTG A5225 Team[¶]

1 Durban University of Technology, Durban, South Africa, 2 Harvard TH Chan School of Public Health, Boston, Massachusetts, and The Biostatistics Center, The George Washington University, Rockville, Maryland, United States of America, 3 Icahn School of Medicine at Mount Sinai, New York, New York, United States of America, 4 Washington University School of Medicine, St Louis, Missouri, United States of America, 5 Social & Scientific Systems, Inc., a DLH Holdings Company, Silver Spring, Maryland, United States of America, 6 Joint Clinical Research Centre, Kampala, Uganda, 7 MOI University Teaching Hospital, Eldoret, Kenya, 8 BJ Medical School, Pune, Maharashtra, India, 9 Research Institute for Health Sciences, Chang Mai, Thailand, 10 University of Zimbabwe, Harare, Zimbabwe, 11 KEMRI Walter Reed Project, Kericho, Kenya, 12 Hospital Nacional Dos de Mayo, Lima, Peru, 13 University of Witwatersrand, Johannesburg, South Africa, 14 University of Southern California, Los Angeles, California, United States of America

☯ These authors contributed equally to this work.
‡ AB, DL, SP, VM, KS, WS, DL, ET, SBF and RAL also contributed equally to this work.
¶ Membership of the ACTG A5225 Team is provided in the Acknowledgments.
* umeshlalloo@gmail.com

## Abstract

### Background

The WHO recommended 1200mg/day of fluconazole (FCZ) in the induction phase of cryptococcal meningitis (CM) in HIV prior to 2018 in regions where amphotericin-B (AMB) was unavailable. A 2-stage AMB-controlled, dose-escalation study to determine the maximum tolerated dose and the safety/efficacy of an induction-consolidation strategy of higher doses FCZ (1200mg-2000mg/day), adjusted for weight and renal function (eGFR)in adults with CM was undertaken.

### Methods

In Stage-1, three induction doses of FCZ (1200mg/day, 1600mg/day and 2000mg/day) were tested in sequential cohortsand compared with AMB in a 3:1 ratio. A particular dose was not tested in Stage 2 if there were significant predetermined safety or efficacy concerns.

**Data Availability Statement:** Due to confidentiality issues and ethical restrictions, study data are available upon request from sdac.data@sdac.harvard.edu with the written agreement of the AIDS Clinical Trials Group.

**Funding:** Research reported in this publication was supported by the National Institute of Allergy and Infectious Diseases of the National Institutes of Health (NIH) under Award Number UM1 AI068634, UM1 AI068636 and UM1 AI106701; clinical trial aidsinfo.nih.gov/clinical-trials/details/NCT00885703.

**Competing interests:** The authors have declared that no competing interests exist.

In Stage-2, the 1200mg dose was excluded per protocol because of increased mortality, and participants were randomised to 1600mg, 2000mg FCZ or AMB in a 1:1:1 ratio.

## Findings

One hundred and sixty eight participants were enrolled with 48, 50, and 48 in the AMB, 1600mg and 2000mg cohorts. The Kaplan Meier proportion for mortality (90% CI) at 10 and 24 weeks for AMB was 17% (10, 29) and 24% (15, 37), compared to 20% (12, 32) and 30% (20, 43) for 1600mg, and 33% (23, 46) and 38% (27, 51) for 2000mg/day FCZ. With the exception of a higher incidence of gastrointestinal side effects in the 2000mg cohort, both induction doses of FCZ were safe and well tolerated. There were no life-threatening changes in electrocardiogram QTc which were similar across all doses of FCZ and AMB. The median (IQR) change in $\log_{10}$ cryptoccal colony forming units (CFU) from week 0 to week 2 was -8(-4.1,-1.9) for AMB; -2.5(-4.0, -1.4) for 1600mg FCZ and -8 (-3.2, -1.0) for 2000mg FCZ. The proportion (90% CI) CSF CM negative at 10 weeks was 81%(71,90) for AMB; 56%(45,69) for 1600mg FCZ and 60%(49,73) for 2000mg FCZ.

## Interpretation

Induction phase weight and renal-adjusted doses of 1600mg and 2000mg/day FCZ for CM were safe and well tolerated except for increased GI side effects in the 2000mg/day dose, and had similar times to achieve CSF sterilization, but took significantly longer than AMB. The WHO recommended 1200mg FCZ was associated with a high mortality. While not statistically significant, mortality was numerically lower in the AMB compared to 1600mg and 2000mg FCZ These data make a case for a phase 3 study of higher doses of FZC.

## Introduction

An estimated 220 000 cases and 180 000 deaths occur annually from cryptococcal meningitis (CM), mostly in sub-Saharan Africa [1]. The USA guideline for management of CM in persons with HIV (PWH) is 2 weeks amphotericin-B (AMB) combined with flucytosine followed by fluconazole (FCZ) at 400mg/day for 8 weeks [2]. Prevention of relapse is FCZ continued at 200mg/day in the maintenance phase for one year and the CD4 count exceeds 100 cells/ulL with complete virologic suppression for at least 3 months on antiretroviral therapy (ART) [2]. Flucytosine is expensive and unavailable in most high burden, resource limited setting and is unregistered throughout Africa [3–5]. AMB is frequently unavailable in many resource limited settings, expensive, complex to administer and associated with serious adverse effects requiring intensive monitoring [2, 3]. Liposomal AMB may be better tolerated but is also unavailable in most resource limited settings due to its cost. The WHO recommendation prior to 2018 was AMB combined with flucytosine or FCZ for 2 weeks induction period, or, in settings where AMB is not available, FCZ combined with flucytosine or high-dose FCZ (1200mg) alone [6]. This was supported by the USA guideline [2, 3]. Doses of FCZ greater than 1200mg/day have demonstrated increased efficacy up to 2000mg/day but only in small numbers of patients [7]. Most guidelines recommend FCZ 400mg/day during the consolidation phase of treatment until week 10, whilst some such as the USA guideline suggest 400–800 mg/day. There is no recommendation for confirming CSF sterility in routine clinical practice to determine the optimal consolidation phase dose of FCZ The newer triazoles have *in vitro* anti-cryptococcal activity,

but clinical data are sparse [8, 9]. Dexamethasone adjunctive treatment was associated with a poorer outcome [10].

An open label phase I/II dose-escalation (Stage-1) and dose-validation (Stage-2) study was planned to investigate the safety, tolerability, and efficacy of an induction-consolidation strategy of high-dose (1200mg-2000mg), daily oral FCZ alone for CM in PWH.

The primary objectives were to determine the maximum tolerated dose (MTD) by comparing the safety and tolerability of induction FCZ regimens of 1200mg, 1600mg, and 2000mg/ day combined with flucytosine where available. The secondary objectives were to compare the clinical outcomes, neurological status, and fungicidal effects of different induction regimens of FCZ during induction through 10 weeks and assess the time to death and the rates of dose limiting toxicity (DLT) and mortality. A concurrently randomized standard of care arm of AMB was included which was revised to allow AMB with flucytosine where available or FCZ per the revised WHO recommendations of December 2011.

## Methods

The study was approved by the USA Food and Drug Administration. In addition, each of the multicentre sites that participated in the study received approval from their local Institutional Review Boards viz: Associacion Civil Impacta–San Miguel Jiron, Lima Peru; Research Institute for Health Sciences, Chiang Mai, Thailand; Joint Clinical Research Centre Mengo, Kampala, Uganda; Walter Reid Project Kenya Medical Research Centre, Kericho, Kenya; USC School of Medicine, Los Angeles, USA; BJ Medical College and Sassoon General Hospital Clinical Trials Unit, Pune, Maharashtra, India; MOI University Teaching Hospital, Eldoret, Rift Valley Kenya; Kilimanjaro Christian Medical Centre, Moshi, Tanzania; University of Witwatersrand CHRU, Helen Joseph Hospital, Johannesburg, South Africa; Durban International Clinical Research Site, Universtiy of KwaZulu-Natal, Durban, South Africa and AIDS Research Unit, Universtiy of Zimbabwe, Harare, Zimbabwe.

The trial was registered with the United States of America National Institutes of Health Clinical Trials.gov identifier NCT00885703.

All participants signed a written informed consent form. A Study Monitoring Committee (SMC) reviewed the study for safety concerns, adverse events and approved escalation to the next dose of FCZ or stopping based on protocol-defined criteria. A core safety team monitored the study safety data monthly.

The study was planned in two stages (Fig 1). Stage-1 was the sequential dose escalation and safety and efficacy stage. In Stage-2 the safe and effective doses were studied in simultaneously enrolled cohorts. Participants were enrolled in consecutive cohorts at each dose level in Stage-1 and randomized to receive FCZ or AMB in a 3:1 ratio They were block randomized without stratification into at each FCZ dose in Stage 1, commencing with 1200mg FCZ and proceeding to the 1600mg and 2000mg dose. This was done sequentially and, per protocol, the next dose was only enrolled once safety was confirmed for the preceding dose by the SMC. In Stage-2, participants were block randomized, without stratification in a ratio of1:1:1 to receive either 1600mg or 2000mg FCZ or AMB. Treatment was planned in 4 steps as follows: Step-1: Induction treatment with high dose FCZ (up to 10 weeks) or AMB (2 weeks)

- Step-2: Induction following early AMB intolerance (only for participants randomised to AMB treatment in Step-1) (FCZ at 400-800mg/day)

- Step-3: Consolidation treatment for AMB arm and for those whose CSF had become negative (FCZ 400mg/day) -up to week 10. Participants whose CSF did not become negative were transitioned to local standard of care

a. Study flow for Stage 1.

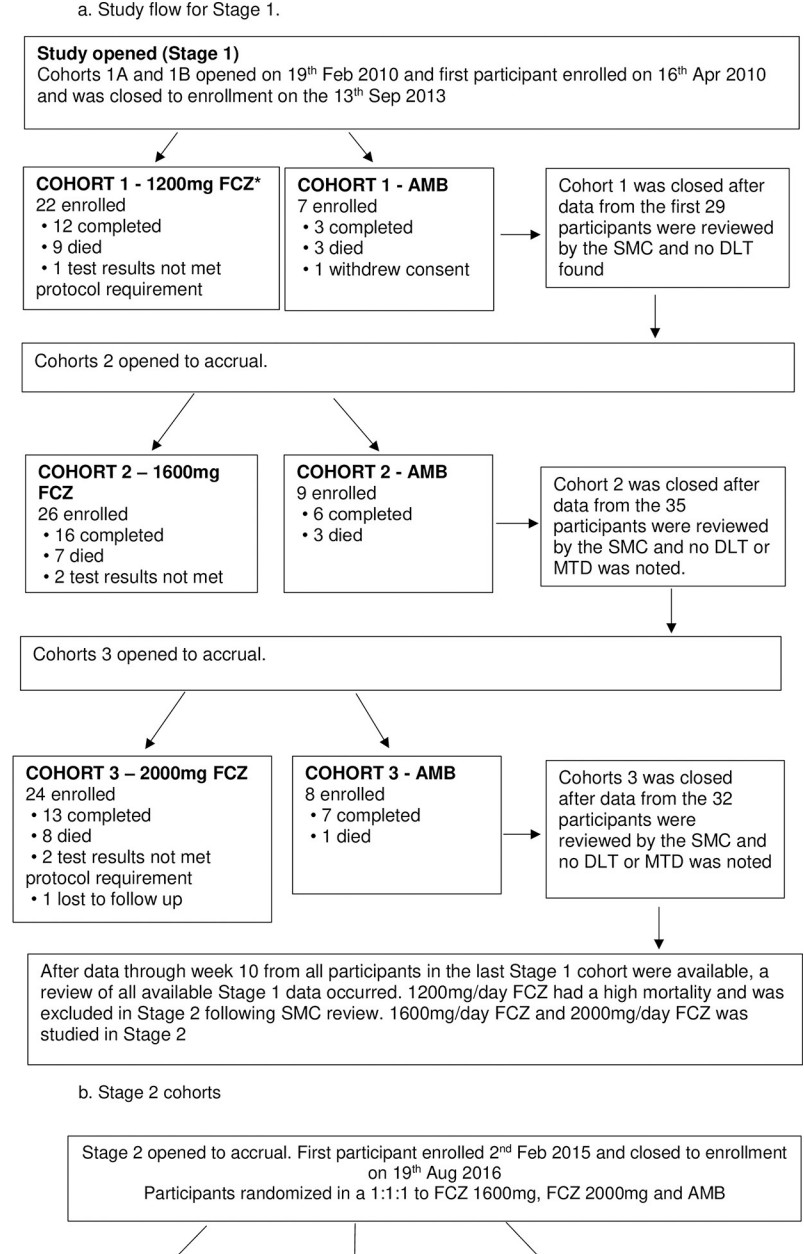

b. Stage 2 cohorts

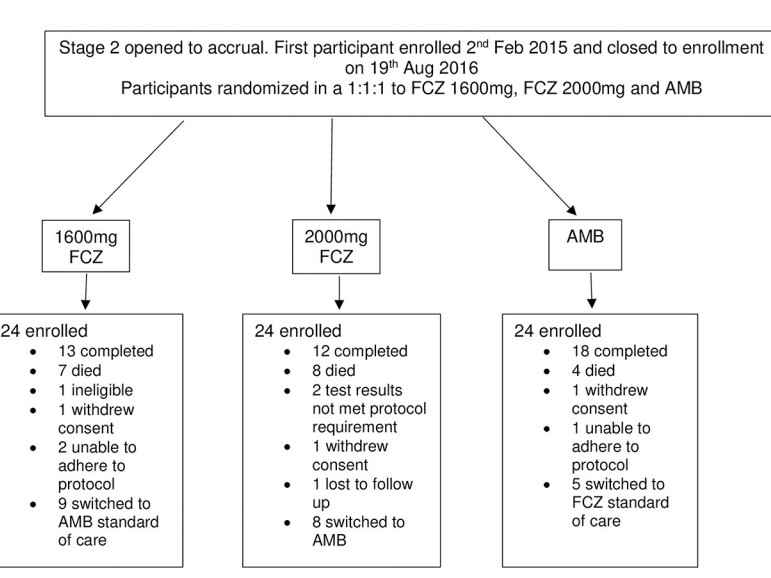

**Fig 1.** a. Study flow for stage 1. b. Stage 2 cohorts. FCZ = fluconazole; AMB = amphotericin B; DLT = dose limiting toxicity; MTD = maximum tolerated dose; SMC = safety monitoring committee. Participants from Stage 1 for each of the 1600mg and 2000mg FCZ and AMB cohorts were combined with the participants from the Stage 2 group for the final safety and efficacy analysis.

- Step-4: Maintenance treatment (FCZ 200mg/day) week 10–24

- 8 participants switched to standard of care AMB.

Participants were followed on study for 24 weeks. The 1200mg/day FCZ arm was excluded per protocol in Stage-2 because, although safe, was associated with a higher mortality.

## Participants

Participants 16 years and older with HIV presenting with the first episode of CM, able to take oral medication and provide signed informed consent were recruited. Nasogastric tube, but not intravenous administration of FCZ was permitted. HIV-1 infection was documented by any licensed rapid HIV test or HIV enzyme or chemiluminescence immunoassay test kit at any time prior to study entry and confirmed within 10 days after study entry. Pregnant and breastfeeding women were excluded. Women of childbearing potential and participating in sexual activity were to use at least two Division of AIDS (DAIDS) approved contraception methods simultaneously. Participants were recruited across 10 international sites.

Cryptococcal meningitis had to be documented by a positive CSF cryptococcal culture, a positive CSF India-ink preparation, or a positive CSF cryptococcal antigen latex agglutination test within 7 days prior to entry. Participants were permitted enrollment with CM that was documented solely by a test result from a referring laboratory. Cryptococcal meningtis had to be confirmed by CSF antigen testing by Day 5 at a DAIDS approved laboratory. Treatment was commenced while awaiting confirmation.

Participants on ART prior to entry were excluded from Stage-1. Participants were permitted to commence ART after at least 4 weeks of CM treatment as per local standard of care [11]. In Stage-2 non-nevirapine containing ART regimens were permitted at entry and throughout the study. Nevirapine was only permitted with FCZ doses < 400mg/day [12]. Rifampicin containing TB treatment was not permitted in the 1200mg FCZ induction cohort. Participants receiving rifampin in the consolidation phase (Step 3 and Step 4) had to take at least 1600mg/day FCZ.

Participants with electrocardiogram (ECG) QTc prolongation greater than 450msec and those receiving concomitant medications documented to prolong QT interval were excluded [13]. The QTc was calculated using the Fredericia formula. Participants with liver enzymes more than 5 times and bilirubin more than 1.5 times the upper limit of normal, absolute neutrophil count < 750/mm$^3$, platelet counts <50 000/mm$^3$ or a hemoglobin < 7g/dL were excluded. Participants with a brain mass lesion that would interfere with the assessment of efficacy or the performance of a lumbar puncture (LP) were excluded.

## Procedures

Clinical status including a comprehensive neurological examination, measurement of the Glasgow Coma Scale (GCS) and mini-mental status was done at entry and repeated 2-weekly or more frequently as indicated and during progression of symptoms.

Safety bloods and ECGs were done at baseline and repeated 2-weekly until week-10 and during progression of symptoms. The study provided LP kits and the opening pressure was

measured and fluid removed for diagnostic purposes and the relief of intracranial pressure. LPs were performed every 2 weeks until the quantitative CSF culture for *Cryptococcus neoformans* was negative, and more frequently as indicated clinically. Quantitative CSF cultures were performed at the clinical research sites' DAIDS approved laboratories that had passed proficiency testing. Quantitative CSF cultures were plated in serial 10-fold dilutions and the dilution with the least colonies, but at least 30 colony-forming units (CFUs) per mL was used to calculate the CFU/mL CSF [14].

## Treatment

Fluconazole provided by Pfizer, was adjusted for body weight-below 60kg: 20mg/kg for 1200mg, 26mg/kg for 1600mg, and 33mg/kg for 2000mg cohorts. The induction phase of FCZ was continued until the CSF culture was negative or to week 10, whichever was earlier. Amphoterecin B was prescribed between 0′7-1mg/kg/day for 2 weeks. Participants who experienced AMB treatment limiting toxicity prior to day 14 or a total of 8.4 mg/kg AMB either remained in Step-1 and were allowed to receive an alternative treatment, such as liposomal AMB, if available, or entered Step-2 and received study-provided FCZ 800 mg daily or the maximum locally approved daily dose. Participants who failed protocol treatment were transitioned to local standard of care. No participant actually received liposomal AMB. Pharmacokinetic (PK) samples were obtained at specified time points from all participants enrolled in FCZ induction cohorts. Whilst it was permitted across the study, only the USA site had access to flucytosine. None of the participants received flucytosine.

The initial FCZ dose was administered as specified, independent of the screening creatinine clearance. Subsequently, FCZ doses were adjusted for clearance rates below 50mL/min, rates between 20-49mL/min, 50% and below 20mL/min 25% of the dose was administered. Participants were followed up for 24 weeks.

## Stopping rules and study discontinuation and treatment failure

Dose limiting toxicity (DLT) in Stage-1 was defined as interruption of FCZ for 3 or more consecutive days because of toxicity and where interruption was initiated prior to day 14. The DAIDS toxicity grading system was used [15]. Three or fewer participants with a toxicity event permitted escalation to the next dose; 4–9 toxicity events would not permit enrolment into the next dose cohort. This would be regarded as the maximum tolerated dose (MTD).; more than 9 toxicity events would indicate that the prior dose would be the MTD. This was assessed after the last participant in the cohort was observed for at least 14 days. A mortality rate of 25% was predicted for CM and enrolment into a cohort would be paused and the SMC consulted if more than 6 early deaths occurred.

Treatment failure for any participant was defined as a positive culture from a CSF sample collected at week 10 and read by week 12, or a positive last CSF culture on study if the participant died or was lost to follow up. If negative CSF culture was obtained prior to week 8, the participant entered Step-3 and received FCZ 400mg/day (or 600mg/day if receiving RIF) until week 10. Some participants in Step-1 did not enter Step-3.

A core safety team consisting of chairs, co-chairs, Clinical Trials Specialist, medical officer, statisticians, and data manager was established. Additionally, A5225 was reviewed by the ACTG Co-infections Subcommitteeand and the SMC. Preplanned SMC reviews were conducted to review safety and tolerability after all participants were followed for at least 14 days for the Stage-1 1200 mg, 1600mg and 2000mg cohorts. A Stage-1 SMC review was conducted after 10 weeks of follow up on participants in the 2000 mg cohort was available to identify which Stage-1 doses would be tested in Stage-2. In Stage-2, SMC reviews were conducted

approximately every six months. Additional safety reviews could be scheduled at the request of the medical officer or other team member. Data from participants in the 1600mg, 2000mg FCZ, and AMB cohorts from Stage-1 were combined with data from the Stage-2 participants in the same cohorts for the final safety, tolerability and efficacy analyses.

## Sample size

The sample size was designed to have a 90% power and a one-sided 0.10 alpha test to detect a change in log10 CFU//ml CSF of about 80% of the standard deviation of the values in Stage 1. Based on this a sample size of 24 was determined and would also permit a dose limiting toxicity rate of >25%. This permitted the identification of the maximum tolerated dose (MTD). The same rationale was used to determine the sample size in Stage 2 once the MTD was identified. In the conduct of Stage 1, the 1200mg FCZ dose was associated with a high mortality and excluded from Stage 2. Up to the maximum dose of 2000mg FCZ there was no MTD, hence the 1600mg and 2000mg FCZ doses were tested in Stage 2.

In Stage-1, substitution was permitted for participants who were HIV negative, CM negative with confirmatory tests in a DAIDS approved laboratory, or who withdrew consent.

The combined Stage-1 and Stage-2 sample size of 48 participants per dose ensured that the width of a 90% CI for proportion culture negative within a treatment cohort was no more than 25%. It also allowed for the detection of a 25% difference in efficacy between 2 doses with 90% power and a one sided 0.10 test. Additionally, with >40 evaluable participants per dose, there is an 87% chance of seeing one or more rare events if the rare or unexpected toxicity event occurs at a rate of 5% or higher.

## Statistical analysis

Participants who received at least one dose of randomised treatment (FCZ or AMB) were included in the safety analysis. Participants whose baseline CSF culture did not grow *Cryptococcus neoformans* were excluded from the efficacy analysis. Participants who died or were lost to follow up prior to a negative CSF culture were counted as culture positive through week 10. Fungicidal efficacy was measured by change in cryptococcal CFU/mL at week 2 and cryptococcal clearance over 10 weeks. Cryptococcal CFU/mL were transformed using a $\log_{10}$ transformation. Mortality was measured over 10 and 24 weeks.

Comparisons for continuous data across the four cohorts used a Kruskal-Wallis test and a Wilcoxon rank sum test for pairwise comparisons. Categorical variables were tested with a Chi-square test. Within arm changes from baseline were tested with a Wilcoxon signed rank test. Time to event data were summarized using the methods of Kaplan-Meier with Greenwood's confidence intervals and log rank tests. Safety results are presented with 95% confidence intervals (CIs). In this phase I/II study, efficacy results are presented with 90% CIs to allow for non-definitive signals of efficacy between FCZ and AMB. No adjustments were made for multiple comparisons.

## Study history

The study was opened to Accrual on February 19, 2010 and the first participant enrolled on April 16, 2010. The first international participant, was enrolled on December 22, 2010. Stage 1 closed to accrual on September 5, 2013. Stage 2 opened to accrual on September 15, 2014 under A5225 Version 3.0 and the first participant enrolled on February 2, 2015. On April 2, 2015, the A5225 team leadership informed the sites of a supply problem with the study-provided FCZ. Due to the limited supply of FCZ that was available for distribution, and in best interest of the study, the A5225 team leadership imposed site-specific screening limits for the

sites registered or in the process of registering for Version 3 of the protocol. Additional FCZ became available in June 2015. However, there was not enough FCZ on hand to complete the study and the team reviewed FCZ drug supply and enrollment, with screening limits adjusted on a site by site basis until additional Fluconazole became available. Stage 2 closed to accrual on August 19, 2016.

Letter of Amendment (LOA) 1 under Version 3 was released on April 20, 2015. A clarification memo under protocol version 3.0 was released on October 12, 2015 in order to clarify treatment options after week 10 treatment failure to help address progression of symptoms. Letter of Amendment 2 for Version 3 was released on October 7, 2016. The LOA was issued to correct the amount of flucytosine allowable prior to study entry; clarify the importance of exercising care in the review and administration of concomitant medications; and to revise the instructions about the follow-up in the event of an abnormal QTc.

The protocol may be accessed at https://clinicaltrials.gov/ct2/show/NCT00885703?term= A5225&rank=1

## Results

The recruitment and entry into Stage-1 and Stage-2 is presented in the consort Fig 1. In the pooled analysis, a total of 168 participants were randomized: 22 to 1200mg FCZ, 50 to 1600mg FCZ, 48 to 2000mg FCZ and, 48 to AMB. Enrolment into 1200mg was stopped at n = 22 following SMC review. The SMC determined that 1200mg FCZ, although safe, demonstrated inferior efficacy based on higher mortality (41%) compared to 30% and 38% reported in the 1600mg and 2000mg FCZ arms, and was was excluded from Stage-2. Participants were enrolled from 10 international sites ("Table 1"). Most were in Uganda, which recruited 34% of the participants. There were no clinically significant differences between participants at this vs. the other sites.

"Table 2" shows the principle demographic, clinical, and laboratory characteristics of the participants at entry in each of the pooled treatment arms. The four cohorts differed on several characteristics; age (p = 0.008), weight (p = 0.01), BMI (p = 0.03) and CrAG titer (p = 0.01). The 1600mg FCZ had the lowest distribution of CrAG titer. The 1200mg FCZ cohort had a higher proportion with a reduced GCS and had a significantly lower BMI compared to the other cohorts.

**Table 1. Accrual of participants by induction dose and ACTG sites.**

| Site | 1200mg FCZ* | 1600mg FCZ* | 2000mg FCZ* | AMB ** | Overall (%) |
|---|---|---|---|---|---|
| Los Angeles,USA: USC | 1 | 0 | 2 | 1 | 4 (2.4) |
| Johannesburg, South Africa: Wits Health Consortium | 0 | 2 | 1 | 1 | 4 (2.4) |
| Durban, South Africa: Enhancing Care Foundation, Durban University of Technology | 1 | 6 | 5 | 10 | 22 (13.1) |
| Lima, Peru: IMPACTA | 0 | 4 | 1 | 1 | 6 (3.6) |
| Kampala, Uganda: Joint Clinical Research Center | 0 | 16 | 24 | 17 | 57 (33.9) |
| Kericho, Kenya: Walter Reed Project | 4 | 4 | 6 | 2 | 16 (9.5) |
| Eldoret, Kenya: AMPATH at MOI University | 4 | 9 | 7 | 6 | 26 (15.5) |
| Harare,Zimbabwe:UZ/UCSF HIV Prevention Trials Unit | 2 | 8 | 2 | 6 | 18 (10.7) |
| Pune, India: BJMC | 5 | 1 | 0 | 3 | 9 (5.4) |
| Chiang Mai, Thailand: Chiang Mai University | 5 | 0 | 0 | 1 | 6 (3.6) |
| TOTAL | 22 | 50 | 48 | 48 | 168 (100) |

*Induction dose of fluconazole

**amphotericin B

**Table 2. Baseline demographic, clinical and laboratory characteristics.**

| TREATMENT GROUP (N) | 1200mg FCZ (N = 22) | 1600mg FCZ (N = 50) | 2000mg FCZ (N = 48) | AMB (N = 48) | p |
|---|---|---|---|---|---|
| Age (yrs) | 40.5 (36, 47) | 33.5 (28, 38) | 36.4 (29, 42) | 36.3 (31, 40) | 0.01* |
| Sex—M | 12 (54.5%) | 26 (52.0%) | 24 (50.0%) | 28 (58.3%) | 0.86** |
| BMI (kg/m$^2$) | 17.2 (15.6, 19.6) | 19.1 (17.1, 21.3) | 19.2 (17.4, 22.5) | 19.7 (17.0, 23.8) | 0.03* |
| Weight (kg) | 46.5 (42.0, 52.2) | 50.3 (44.0, 63.0) | 54.8 (47.5, 59.1) | 54.0 (48.5, 62.0) | 0.01* |
| Height (cm) | 163.0 (155.0, 169.0) | 162.7 (156.0, 171.1) | 166.5 (158.5, 172.5) | 167.3 (162.0, 171.0) | 0.10* |
| CD4 count (cell/mm$^3$) | 20 (9, 55) | 25 (8, 75) | 28.0 (9, 63) | 26 (9, 58) | 0.82* |
| HIV RNA (log10) (copies/mL) | 5.3 (5.1, 6.0) | 5.3 (5.0, 5.8) | 5.3 (4.7, 5.8) | 5.3 (4.4, 5.7) | 0.39* |
| On ART [†] | 0 (0.0%) | 8 (16.0%) | 12 (25.0%) | 11 (22.9%) | - |
| OIs >0 | 10 (45.5%) | 18 (36.0%) | 20 (41.7%) | 20 (41.7%) | 0.87** |
| GCS <15 | 5 (22.7%) | 5 (10.0%) | 5 (10.6%) | 1 (2.1%) | 0.06** |
| Mini-Mental Status | 26 (23, 26) | 26 (21, 26) | 26 (20, 26) | 26 (24, 26) | 0.85* |
| Opening Pressure (mm H$_2$O) | 200 (105, 300) | 200 (150, 258) | 210 (150, 420) | 260 (140, 420) | 0.17* |
| Albumin (g/dL) | 2.9 (2.6,3.3) | 3.2 (3.0,3.4) | 3.2 (2.7,3.6) | 3.9 (2.7,3.6) | 0.16* |
| ALT (SGPT) | 19.3 (11.0,22.8) | 34.9 (14.9,33.0) | 29.4 (14.4,35.0) | 26.3 (13.5,33.5) | 0.15* |
| Creatinine clearance (mL/min) | 95.6 (79.5,112.1) | 92.9 (75.0,105.7) | 95 (69.4,118.5) | 96.1 (80.0,116.6) | 0.60* |
| CrAG titer | 756 (256, 8192) | 153.5 (32, 1024) | 512 (32, 4063) | 1024 (160, 4096) | 0.01* |
| Crypto CFU (log10) | 5.2 (4.2, 6.0) | 5.0 (4.0, 5.4) | 4.4 (3.3, 5.4) | 4.7 (3.4, 5.3) | 0.27* |

Note: Median (IQR) for continuous, count (%) for categorical [†] Stage 1 did not include participants on ART at baseline, while Stage 2 did.

* Wilcoxon rank sum test

** Chi-square test

There were 10/22 participants in the 1200mg FCZ, 7/26 in the 1600mg, 11/24 in the 2000mg FCZ cohort and 10/24 in the AMB cohort in the Stage 1 with non CM opportunistic infections and AIDS defining conditions. None of these were clinically severe and there was no significant difference in the frequency between the cohorts. Likewise, in Stage 2 there were 11/24 and 9/24 participants in the 1600mg and 2000mg FCZ and 10/24 in the AMB cohorts with non CM opportunistic infections and AIDS defining conditions. There was also no significant difference among the cohorts. The actual non CM opportunistic and AIDS defining conditions are shown in "Table 3". Candidiasis, oropharyngeal, oesophageal or vaginal accounted for the majority of opportunistic infections.

Rifampicin was permitted in Stage 2 to facilitate recruitment in high TB incident regions. Rifampicin was commenced in 4 participants in the induction phase of treatment in the FCZ cohort: 1 in the 1600mg cohort commenced on Day 11 and survived. There were 3 in the 2000mg cohort: 1 commenced on day 2, and died on day 80; 1 commenced on day 3 and survived; and 1 commenced on day 10 and died on day 80. No dose adjustments were planned for participants on rifampicin. Excluding these participants from the analysis made no difference to the statistical outcome.

Fourteen participants who received at least one dose of study medication (FCZ or AMB) were included in the safety but not efficacy analysis: 12 negative for *Cryptococcus neoformans* (11 culture negative and 1 non-confirmed at a DAIDS approved laboratory), 1 commenced anti-TB treatment, and 1 deteriorated clinically and could not take medication enterally. Anti-TB treatment was an exclusionary criterion in Stage 1. Two had negative CSF cultures in the AMB cohort and 10 distributed among the FCZ cohorts. One participant in the AMB cohort was included in the safety and efficacy analyses despite having been enrolled with a prolonged QTc interval which was an exclusionary criterion. The efficacy analysis included 134

**Table 3. Baseline non-cryptococcal meningitis opportunistic infections and AIDS-defining conditions by Stage and treatment.**

| Diagnosis Code | Stage 1: 1200mg fluconazole | Stage 1: 1600mg fluconazole | Stage 1: 2000mg fluconazole | Stage 1: ampho-terecin B | Stage 2: 1600mg fluconazole | Stage 2: 2000mg fluconazole | Stage 2: ampho-terecin B | Total |
|---|---|---|---|---|---|---|---|---|
| **Malaria—confirmed** | 1 | 0 | 0 | 0 | 0 | 0 | 0 | 1 |
| **Toxoplasmic encephalitis—probable** | 0 | 0 | 0 | 0 | 0 | 1 | 0 | 1 |
| **Disseminated cryptococcosis—confirmed** | 1 | 0 | 0 | 1 | 0 | 0 | 0 | 2 |
| **Disseminated cryptococcosis—probable** | 1 | 0 | 0 | 0 | 0 | 0 | 0 | 1 |
| **Oral/oropharyngeal candidiasis, specify oral or oropharyngeal-confirmed-pseudomembranous candidiasis** | 0 | 0 | 0 | 0 | 1 | 0 | 0 | 1 |
| **Oral/oropharyngeal/oesophageal,- oral or oropharyngeal-probable-pseudomembranous/erythematous** | 3 | 4 | 7 | 6 | 3 | 3 | 3 | 29 |
| **Vulvovaginal candidiasis—probable** | 0 | 0 | 1 | 0 | 0 | 0 | 0 | 1 |
| **Fungal nail infections—probable** | 0 | 2 | 0 | 0 | 0 | 0 | 0 | 2 |
| **Pulmonary tuberculosis-clinical diagnosis/probable** | 0 | 0 | 0 | 2 | 0 | 1 | 1 | 4 |
| **Pulmonary tuberculosis (tb)—confirmed** | 0 | 0 | 0 | 1 | 1 | 1 | 0 | 3 |
| **Mucocutaneous herpes simplex—probable** | 0 | 0 | 0 | 0 | 1 | 0 | 0 | 1 |
| **Herpes labialis** | 0 | 0 | 1 | 0 | 0 | 0 | 0 | 1 |
| **Acute gastrointestinal/diarrheal syndrome** | 1 | 0 | 0 | 0 | 0 | 1 | 0 | 2 |
| **Persistent diarrhoea** | 0 | 1 | 0 | 0 | 0 | 0 | 0 | 1 |
| **Angular cheilitis—probable** | 0 | 0 | 1 | 0 | 0 | 0 | 0 | 1 |
| **Bacterial pneumonia—probable / clinical** | 2 | 0 | 1 | 0 | 1 | 0 | 0 | 4 |
| **Salmonella sepsis (non-typhoid)–confirmed** | 1 | 0 | 0 | 0 | 0 | 0 | 0 | 1 |
| **Bacterial infection-deep tissue/other normally sterile site-confirmed** | 0 | 0 | 2 | 0 | 0 | 0 | 0 | 2 |
| **Eye, ear, nose disease** | 1 | 0 | 0 | 0 | 1 | 1 | 0 | 3 |
| **Genitourinary—renal system disease/disorder, other** | 1 | 0 | 0 | 0 | 0 | 0 | 0 | 1 |
| **CNS disease/disorder, other** | 1 | 0 | 0 | 0 | 0 | 0 | 0 | 1 |
| **Peripheral nerve disease/disorder, other** | 0 | 0 | 1 | 0 | 0 | 0 | 0 | 1 |
| **Hematologic disease (other than clotting disorder)** | 3 | 1 | 1 | 2 | 2 | 2 | 0 | 11 |
| **Dermatologic—skin disease/disorder, other** | 3 | 2 | 2 | 0 | 3 | 0 | 2 | 12 |
| **STD—venereal disease, other** | 0 | 0 | 0 | 0 | 0 | 0 | 1 | 1 |
| **Neurologic system disease/disorder, other** | 0 | 1 | 0 | 0 | 0 | 0 | 0 | 1 |
| **Wasting syndrome—clinical diagnosis only** | 0 | 1 | 0 | 0 | 0 | 0 | 0 | 1 |
| **STD–warts** | 0 | 0 | 1 | 0 | 0 | 0 | 0 | 1 |

participants: 45, 43, and 46 in 1600mg, 2000mg and AMB cohorts. All results for time to negative culture and mortality are in the efficacy population.

At baseline, the treatment groups did not differ in *Cryptococcus neoformans* CFUs, CD4 counts, or HIV viral load (all p>0'25). Of the 16 (9'6%) with a Glasgow Coma Scale (GCS) of

less than 15, 15 were assigned FCZ (p = 0.06). All except one participant in the 2000mg cohort was hospitalised at enrolment.

Doses of FCZ were adjusted for weight in 70%, and 71% of participants in the 1600mg, and 2000mg cohorts. Actual doses ranged from 1000mg to 1600mg in the 1600mg cohort and 1000mg to 2000mg in the 2000mg cohort. The creatinine clearance at entry was similar across the cohorts. In the AMB cohort, there was a significant drop in creatinine clearance by week 2, (Median (IQR) -25.5 mL/min (-37.8, -12.3) Wilcoxon signed rank p = 0'001) but it returned to baseline value by 10 weeks.

Thirty-one participants (16%, 25%, and 23% in 1600mg, 2000mg FCZ, and AMB) were on ART prior to study entry in Stage-2. ART was not permitted in Stage 1. ART was commenced in 88 (55%, 54%, 44%, and 58% in 1200mg, 1600mg, 2000mg, FCZ and AMB) after a median of 34 days (range 1–88).

Fig 2 shows the Kaplan-Meier plot for the time to negative culture in all participants in the efficacy analysis. The data for the 1200mg cohort is included in the plot for comparison. The Kaplan-Meier proportions of *Cryptococcus neoformans* CSF negative at week 10 (90% Greenwood's CI) were 56% (45, 69), 60% (49, 73), and 81% (71, 90) for the 1600mg, 2000mg FCZ, and AMB cohorts. The pairwise comparisons for the 1600mg and 2000mg FCZ arms to the AMB arm were p = 0'06 and 0'02 and are statistically significant at the 0.10 threshold. The 50[th]

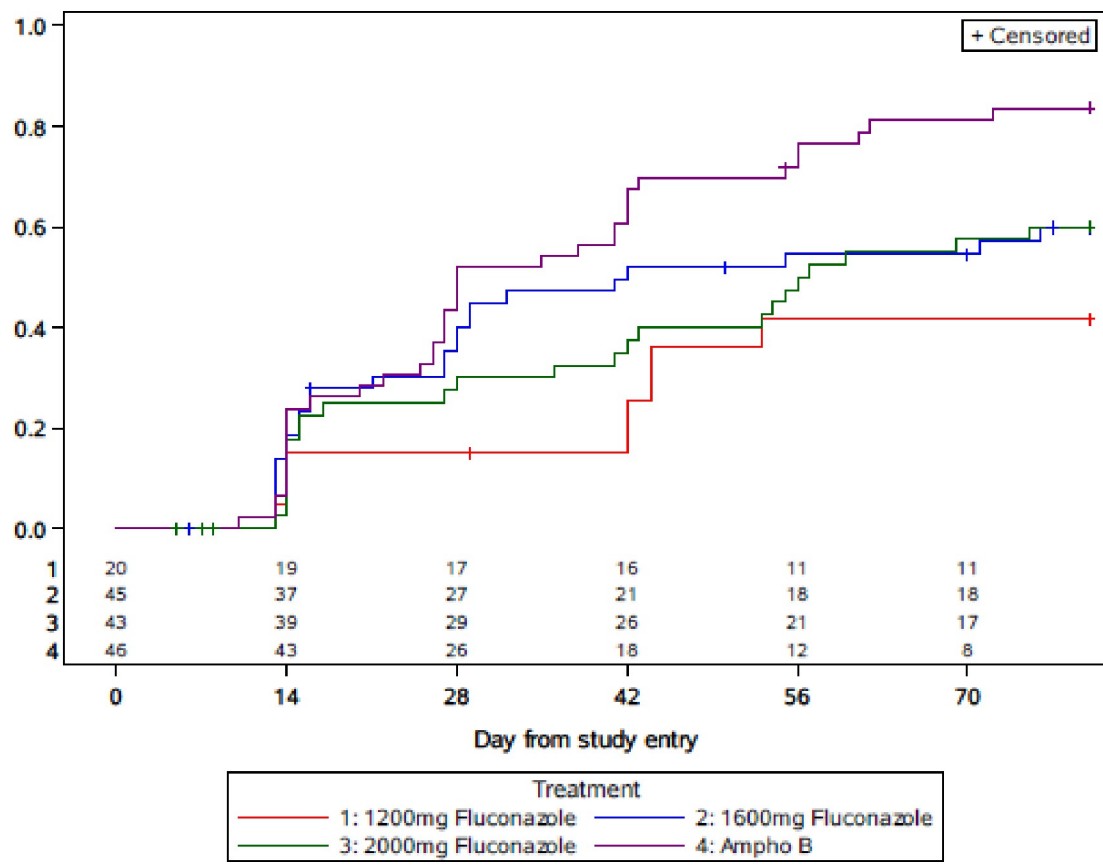

**Fig 2. Time to negative CM culture.** * In the 1600mg FZ arm, 4/45 subjects switched to Ampho B and culture converted by week 10, An additional 5/45 started ampho B and subsequently died. In the 2000mg FZ arm 5/43 subjects switched to Ampho B and culture converted by week 10. An additional 3/43 started ampho B and subsequently died. In the Ampho B arm, 1/46 subjects switched to Fluconazole and culture converted by week 10.

**Table 4. Efficacy population: Cryptococcal CFU (Log10) at Weeks 0 and 2.**

| Median (IQR) Crytpo CFU log10 | 1200mg FCZ | 1600mg FCZ | 2000mg FCZ | AMB | Kruskal WallisP |
|---|---|---|---|---|---|
| | Week 0 (N = 20) | Week 0 (N = 45) | Week 0 (N = 43) | Week 0 (N = 46) | |
| | Week 2 (N = 15) | Week 2 (N = 36) | Week 2 (N = 37) | Week 2 (N = 40) | |
| Week 0 | 5.2 (4.2, 6.0) | 5.0 (4.0, 5.4) | 4.4 (3.3, 5.4) | 4.7 (3.4, 5.3) | 0.26 |
| Week 2 | 4.2 (1.9, 4.5) | 2.1 (0.0, 2.7) | 2.0 (0.0, 3.2) | 1.0 (0.0, 2.2) | 0.013 |
| Change from Week 0 to Week 2 | -1.5 (-2.9, -0.9) | -2.5 (-4.0, -1.4) | -1.8 (-3.2, -1.0) | -2.8 (-4.1, -1.9) | 0.019 |

All p-values are calculated using the Kruskal-Wallis test for differences across the four treatment arms.

centile time (90% CI) to *Cryptococcus neoformans* culture conversion to negative was 42 (27, 74) days, 55 (32, 75) days, and 28 (26, 42) days for 1600mg, 2000mg FCZ, and AMB.

The median $\log_{10}$ CFU (IQR) quantitative CSF culture results at study entry were 5'0 (4'0, 5'4), 4'4 (3'3, 5'4), and 4'7 (3'4, 5'3) for 1600mg, 2000mg FCZ, and AMB ("Table 4"). The median (IQR) change from entry to week 2 of the CSF cfu/mL was -2'5 $\log_{10}$ (-4'0, -1'4), -1'8 $\log_{10}$ (-3'2, -1'0), and -2'8 $\log_{10}$ (-4'1, -1'9) 1600mg, 2000mg FCZ, and AMB. The decrease in *Cryptococcus neoformans* from CSF in the AMB arm did not differ from the 1600mg arm (p = 0'40) and was significantly greater than the 2000mg FCZ arm (p = 0'02).

The Kaplan-Meier proportion for mortality (90% CI) in the efficacy population at 10 and 24 weeks for AMB was 17% (10, 29) and 24% (15, 37), compared to 20% (12, 32) and 30% (20, 43) for 1600mg, and 33% (23, 46) and 38% (27, 51) for 2000mg FCZ (Fig 3). Data for the

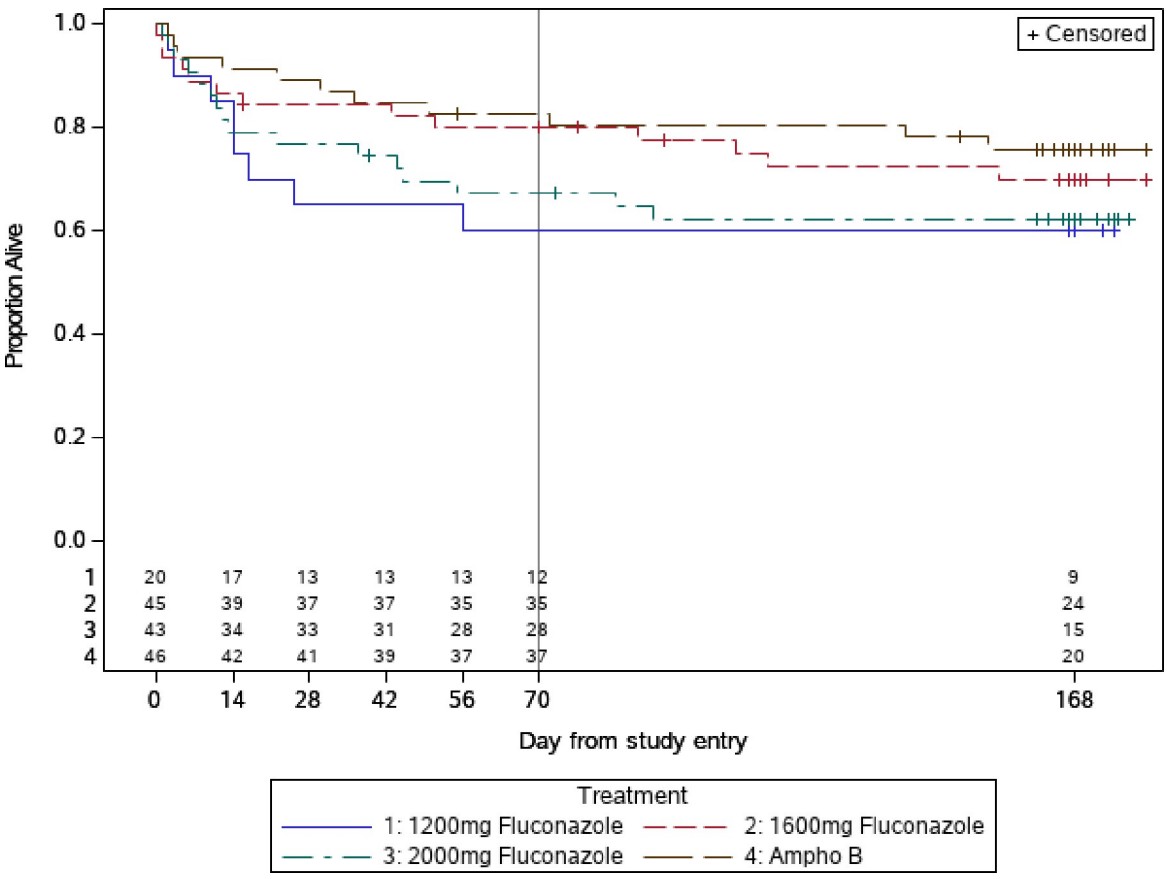

**Fig 3. Time to death (safety population).**

**Table 5. Pairwise comparisons of FCZ with AMB.**

| Treatment Arm | KM Proportion CM Negative by week 10 (90% CI) | p-value comparing to AMB arm | KM Proportion who died by week 10 (90% CI) | p-value comparing to AMB arm | KM Proportion who died by week 24 (90% CI) | p-value comparing to AMB arm |
|---|---|---|---|---|---|---|
| **1200mg FCZ** | 0.45 (0.29, 0.65) | 0.002 | 0.40 (0.25, 0.60) | 0.07 | 0.40 (0.25, 0.60) | 0.14 |
| **1600mg FCZ** | 0.56 (0.45, 0.69) | 0.06 | 0.20 (0.12, 0.32) | 0.89 | 0.30 (0.20, 0.43) | 0.53 |
| **2000mg FCZ** | 0.60 (0.49, 0.73) | 0.02 | 0.33 (0.23, 0.46) | 0.15 | 0.38 (0.27, 0.51) | 0.14 |
| **AMB** | 0.81 (0.71, 0.90) | - | 0.17 (0.10, 0.29) | - | 0.24 (0.15, 0.37) | - |

*All pairwise p-values calculated using the log-rank test

1200mg FCZ is shown in the Fig 3 for reference. These results show that most deaths occurred in the first 10 weeks. In pairwise comparisons, neither FCZ 1600mg or 2000mg differed statistically from the AMB arm (all p≥0'14, "Table 5") at week 10 or week 24.

The presumed causes of death were classified according to the ACTG Appendix 60 and are shown in Table 6. CM accounted for the majority (48%) of deaths. Progression of symptoms, defined as clinician reporting of progression of symptoms, death, discontinued FCZ and started AMB, or a positive *Cryptococcus neoformans* culture at week 10, is shown in the Kaplan-Meier plot (Fig 4).

"Table 7" presents the important adverse events (DAIDS Grade 2 or worse) excluding deaths that occurred during the study period by treatment cohort [15]. A high proportion of participants had Grade ≥2 toxicity/signs/symptoms in all cohorts reported as percent (95% CI): 100% (85, 100), 78% (65, 87), 90% (78, 95), and 75% (61, 85) for the 1200mg, 1600mg, 2000mg FCZ, and AMB cohorts. Nausea and vomiting of Grade 2 or higher intensity was reported in 7 of 50 participants (6 with vomiting only, 1 with nausea and vomiting) in the

**Table 6. Causes of death.**

| Cause* | 1200mg FCZ (N = 9) | 1600mg FCZ (N = 14) | 2000mg FCZ (N = 16) | AMB (N = 11) | Overall (N = 50) |
|---|---|---|---|---|---|
| **Cryptococcal meningitis—confirmed** | 4 | 7 | 7 | 6 | 24 |
| **Cryptococcal meningitis—probable** | 0 | 1 | 0 | 0 | 1 |
| **Disseminated cryptococcosis—confirmed** | 2 | 1 | 0 | 0 | 3 |
| **CNS disease/disorder, other** | 1 | 0 | 0 | 0 | 1 |
| **Extra pulmonary tuberculosis—probable** | 0 | 0 | 1 | 0 | 1 |
| **Genitourinary—renal system disease/disorder, other** | 0 | 0 | 1 | 0 | 1 |
| **Malaria–probable** | 0 | 0 | 0 | 1 | 1 |
| **Neurologic system disease/disorder, other** | 0 | 0 | 0 | 1 | 1 |
| **Bacterial sepsis/catheter related bacteremia/sepsis -confirmed** | 0 | 1 | 0 | 1 | 2 |
| **Bacterial sepsis/catheter related bacteremia/sepsis -clinical diagnosis only** | 0 | 0 | 1 | 0 | 1 |
| **Bacterial pneumonia—probable** | 1 | 0 | 0 | 0 | 1 |
| **Probable pneumonia and/or etiology unknown** | 1 | 0 | 2 | 0 | 3 |
| **Pulmonary—respiratory disease/disorder, other** | 0 | 1 | 2 | 0 | 3 |
| **Pulmonary—respiratory failure** | 0 | 1 | 0 | 0 | 1 |
| **Pulmonary embolus—probable** | 0 | 0 | 1 | 0 | 1 |
| **Other event not listed in appendix 60*, non-HIV associated** | 0 | 0 | 1 | 0 | 1 |
| **No information available** | 0 | 2 | 0 | 2 | 4 |

*Defined by AIDS Clinical Trials Group (ACTG) Appendix 60

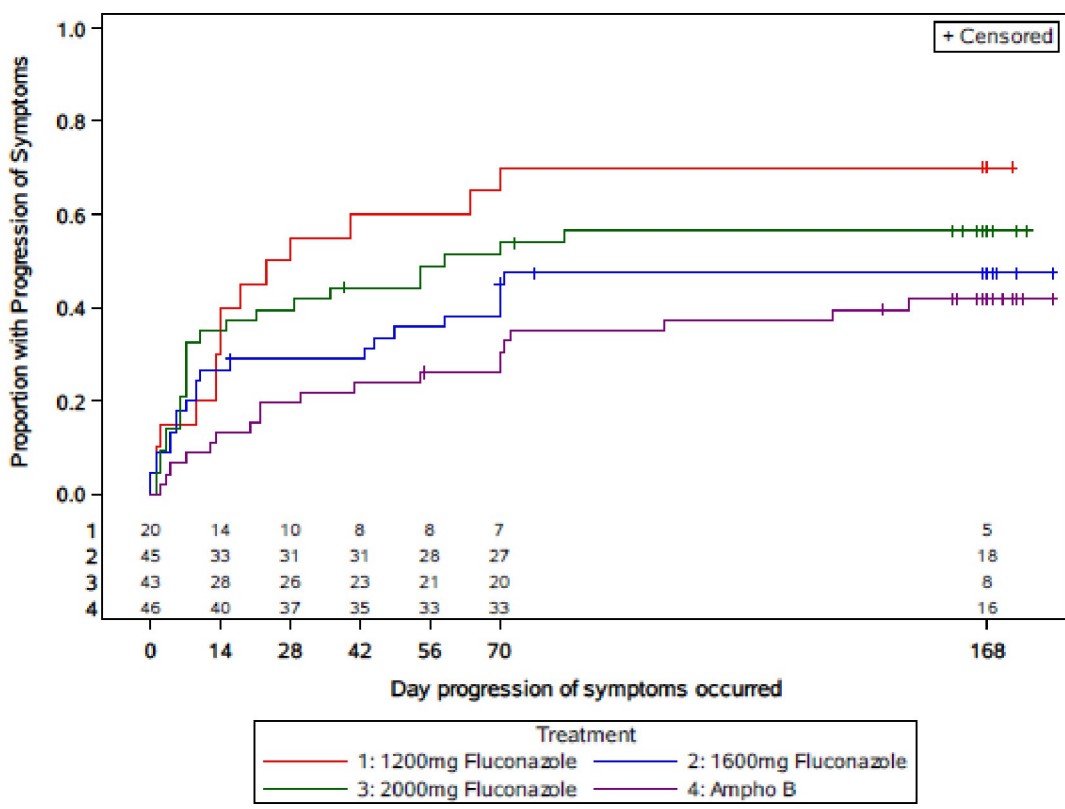

**Fig 4. Time to progression of symptoms through end of study.**

1600mg and 16 of 48 (13 with vomiting, 1 with nausea, 2 with vomiting and nausea) in 2000mg cohort (p<0'03 for the pairwise comparison).

The median (IQR) duration of hospitalisation was 18 (13, 28) days and was similar across cohorts. There was no significant difference in the subsequent hospitalisation by cohorts: 11 (22%), 7 (15%) and 11(23%), 1600mg, 2000mg FCZ and AMB. Two participants developed immune reconstitution inflammatory syndrome (IRIS). One was in the 1200mg FCZ cohort, who commenced antiretroviral treatment on day 63 and manifested IRIS on day 99. The other was in the AMB cohort and commenced antiretroviral treatment on day 26 and manifested IRIS on day 38.

QT prolongation is a specific adverse event of interest because of the high doses of FCZ and the blackbox warning for QTc with high dose FCZ and was specifically monitored with periodic ECGs. "Table 8" shows the DAIDS QTc grade of or change from baseline for all cohorts. Overall 45%, 32%, 44%, and 33% had QTc prolongation of Grade ≥2 for 1200mg, 1600mg, 2000mg FCZ, and AMB. None were deemed life-threatening (QTc>500msec). QTc prolongation was similar across all cohorts including AMB for all grades of severity. Creatinine clearance measured at Days 4 and 7 and week 2 showed a significant difference across all cohorts (p = 0'001) with a lower clearance for AMB and 1200mg FCZ. Grade 4 renal toxicity was observed in 2/120 (2%) participants who received FCZ vs. 4/48 (8%) in those receiving AMB. A Grade 4 drop in haemoglobin was found in 6/120 (5%) receiving FCZ vs. 4/48 (8%) in those receiving AMB; Grade 4 hyponatraemia was observed in 13/120 (11%) receiving FCZ vs. 1/48 (2%) in those receiving AMB; and Grade 4 hypokalaemia was observed in 2/120 (3%) receiving FCZ vs. 4/48 (8%) in those receiving AMB. One participant in the AMB had adrenal insufficiency and was prescribed fludrocortisone and glucocorticosteroids. Several participants with

**Table 7. List of grade 2 or worse adverse events through end of study.**

| | 1200mg FCZ N = 22 | 1600mg FCZ N = 50 | 2000mg FCZ N = 48 | AMB N = 48 | p |
|---|---|---|---|---|---|
| **Any Grade > = 2** | 22 (100%) | 39 (78%) | 43 (90%) | 36 (75%) | 0.02 |
| **Any Neurological** | 16 (73%) | 25(50%) | 34 (71%) | 28 (58%) | 0.18 |
| • headaches | 12 (55%) | 22 (44%) | 29 (60%) | 26 (24%) | |
| • worsening level of consciousness/lethargy | 4 (18%) | 1 (2%) | 4 (8%) | 0 (0%) | |
| • confusion/difficulty concentrating | 6 (27%) | 3 (6%) | 8 (17%) | 3 (6%) | |
| **Any gastrointestinal*** | 6 (27%) | 12 (24%) | 22 (46%) | 13 (27%) | 0.21 |
| • nausea | 1 (5%) | 1 (2%) | 3 (6%) | 2 (4%) | |
| • vomiting | 2 (9%) | 7 (14%) | 15 (31%) | 8 (17%) | |
| **Any circulatory/cardiac** | 11 (50%) | 17 (34%) | 23 (48%) | 18 (38%) | 0.23 |
| • QTc prolongation | 10 (45%) | 16 (32%) | 21 (44%) | 16 (33%) | |
| **Any haematology** | 10 (45%) | 24 (48%) | 20 (42%) | 27 (56%) | 0.26 |
| • neutropaenia | 4 (18%) | 16 (32%) | 13 (27%) | 15 (31%) | |
| • anaemia | 7 (32%) | 10 (20%) | 9 (19%) | 22 (46%) | |
| • thrombocytopaenia | 6 (27%) | 6 (12%) | 5 (10%) | 7 (15%) | |
| **Any hepatic** | 4 (18%) | 7 (14%) | 5 (10%) | 8 (17%) | 0.39 |
| ALT (SGPT)* | 3 (14%) | 5 (10%) | 4 (8%) | 3 (6%) | |
| **Renal** (creatinine) | 4 (18%) | 5 (10%) | 5 (10%) | 14 (29%) | 0.05 |
| Hypo/hyperkalaemia | 10 (45%) | 3 (6%) | 7 (15%) | 22 (46%) | |
| Hypo/hypernatraemia | 11 (50%) | 23 (46%) | 22 (46%) | 16 (33%) | |

Any (sign/symptom or lab) includes more levels than listed here

Any gastrointestinal includes: Appetite loss/decreased/anorexia, Constipation, Diarrhea/loose stools, Nausea, Vomiting, Gastrointestinal dysfunction.

All p-values are calculated using the Kruskal-Wallis test for differences across the four treatment arms.

hyponatremia and hyperkalaemia received a glucocorticosteroid: 9/25 FCZ 1600 mg, 8/23 FCZ 2000mg, and 9/19 AMB.

Functional status assessed as ability to work 6 weeks prior to study entry, at time of enrolment, at week 10 and at week 24 did not differ across the cohorts. Likewise, the functional status did not differ in any cohort at any visit. Of those who were alive and for whom the data were available 60% (18/30), 74% (17/33), and 68% (22/33) were back at work or involved in full time activity at week 24 in the 1600mg, 2000mg FCZ, and AMB cohorts. (p = 0'7).

## Discussion

This study has shown that weight based FCZ higher than 1200mg and up to 2000mg/day in the induction phase of the first episode of CM in PWH is safe and effective. Amphotericin B

**Table 8. Maximum DAIDS QTc* Grade of Visit or change from baseline.**

| DAIDS GRADE QTc* | 1200mg FZC n = 22 | 1600mg FCZ n = 50 | 2000mg FCZ n = 48 | AMB n = 48 | TOTAL n = 168 |
|---|---|---|---|---|---|
| **1. normal** | 5 (23%) | 16 (32%) | 14 (29%) | 12 (25%) | 47 (28%) |
| **2. mild** | 7 (32%) | 18 (36%) | 13 (27%) | 20 (42%) | 58 (35%) |
| **3. moderate** | 5 (23%) | 11 (22%) | 14 (29%) | 10 (21%) | 40 (24%) |
| **4. severe** | 5 (23%) | 5 (10%) | 7 (15%) | 6 (12.5%) | 23 (14%) |

*QTc calculated using the Frederica equation.

Note: A change from baseline can result in a graded QTc while the QTc result at the visit is still in the normal range. There was no significant difference between groups using the Kruskall Wallis test.

was associated with better fungicidal activity than FCZ at any dose tested. Survival was improved but not statistically significantly in the AMB cohort compared with the 1600mg and 2000mg FCZ arms. We demonstrated that doses of FCZ higher than the 1200mg/day as recommended by the WHO and USA Guidelines had better CSF sterilisation at week 10 (2,3). From a study in Malawi, it was also shown FCZ 1600mg/day and 2000mg/day alone in the induction phase was more fungicidal compared to FCZ 1200mg/day combined with flucytosine as reported [4].

Our study was conducted simultaneously across 10 international sites ("Table 1") and had AMB as a contemporaneous control. The strengths are the inclusion of stringent criteria for CM diagnosis, standardization of intracranial hypertension management and procedures for quantitative culture of *Cryptococcus neoformans* in multinational resource limited settings with the highest burden of AIDS.

This study was conducted in 2 stages and per protocol Stage 1 was a dose determining strategy for stage 2. Based on predetermined criteria the 1200mg FCZ arm was stopped due to higher mortality. It is possible that the lower BMI in this cohort may partly explain the higher mortality. There was no statistical difference in clinical severity and demographic criteria between the 1200mg, and the 1600mg and 2000mg cohorts in Stage 1.

The bioavailability of oral FCZ has been studied in humans from 50 to 400mg daily; enteral absorption of FCZ is 90%, not affected by food, and demonstrates linear pharmacokinetics within this dose range. Based on the dose response within the dose range in this study, the linearity is likely to continue at least up to a dose of 2000mg. The CSF penetration of FCZ is presumed to be linear as well explaining the CM dose response effect up to 2000mg/day, although no CSF drug levels were analysed in the present report [16–18]. A meta-analysis suggested modest improvements in both CSF sterility and mortality outcomes with escalating dosage of FCZ up to 1200mg but inadequate for sterility [18]. This meta-analysis demonstrated that doses of FCZ up to 1200mg daily is inadequate for induction treatment for CM supporting the rationale of our study.

Rifampicin may reduce the bioavailability of FCZ and was only permitted in Stage 2 of the protocol in the 1600mg and 2000mg FCZ cohorts. A pharmacokinetic study of 400mg FCZ in the consolidation phase of CM following 2 weeks of AMB, showed a 39% increase in the elimination constant of serum FCZ with concomtitant rifamipicin at 600mg daily [19]. This did not alter the outcomes for CM. Our study did not adjust FCZ doses of the doses and excluding participants on rifampicin did not make any significant didference. We propose that a larger phase 3 study of higher doses of FCZ should take this into account.

The duration of the present study was almost 7 years and reflected several challenges in the implementation and conduct across multiple resource limited international sites. These included delays in obtaining IRB and country approvals, determination of participation at national level by government agencies, training of laboratory personnel to conduct quantitative cultures with proficiency testing, delays in approval of laboratories to Division of AIDS of the USA National Institutes of Health standards, availability of AMB at sites, widespread implementation of ART programs, introduction of screening for serum cryptococcal antigen, and implementation of FCZ prophylaxis for serum cryptococcal antigen positive persons initiating ART [20–22]. Per protocol inherent delays occurred because the SMC needed to meet after Day 14 data from each Stage-1 cohort's last participant before the next dose could be opened for enrolment, and Stage-2 could only commence after a protocol amendment that was based on the outcome of an SMC review that was convened after week 10 data from the last participant in the 2000mg/day in Stage was available.

The mortality in our study was much lower than that reported in a previous study comparing 1200mg FCZ to 1200mg FCZ combined with flucytosine [5]. In that study, the mortality with addition of flucytosine was reduced from 58% to 43% [5]. In our study, the mortality in

the 1200mg FCZ cohort was 41%. The mortality was 30% and 38% in the 1600mg and 2000mg/day respectively at 24 weeks. Only the USA site had access to flucytosine that was permitted in our study, but no participant received flucytosine. It is likely that the lower mortality in our study is due to the improved standard of care implemented because of upscaling of resource limited settings through the DAIDS ACTG program of research and the inclusion of participants with a relatively higher GCS [1]. A Cochrane meta-analysis of several treatment combinations for CM in HIV showed no significant difference in mortality with FCZ compared to AMB combined with FCZ [23]. There were no studies with significant numbers of participants with FCZ doses higher than 1200mg daily in the induction phase of treatment.

The WHO recommended dose of FCZ 1200mg daily in the induction phase was shown to be more effective than 800mg in a study in Uganda that enrolled 30 participants in each group sequentially [24]. Notably the Uganda study had no AMB controls and flucytosine was not used. Whilst the 1200mg dose had greater fungicidal activity, there was no mortality difference between the 2 groups. The mortality was 30% and 54% at 2 and 10 weeks respectively whilst the mortality in our study was much lower and the difference may well be attributed to enrolling patients with more severe disease in the Uganda study (47% with reduced level of consciousness) and/or the higher doses of FCZ employed in our study. A prospective study from Ethiopia showed a mortality of 68% (23/34) in CM using 1200mg FCZ induction doses [25]. A mortality of 55% (26/47) was also observed in a prospective study conducted in Malawi [26]. These observations confirm that the 1200mg induction dose of FCZ for CM is inadequate.

Fluconazole is known to prolong the ECG QTc interval at high doses ($\geq$1200mg/day). This study recorded a high incidence of QTc prolongation that was not associated with increased mortality or life-threatening cardiac arrhythmias. This was likely due to frequent ECG monitoring and avoiding concomitant medications known to prolong the QTc. It is notable that QTc prolongation across grades of severity from mild to severe was similar in all FCZ cohorts and in the AMB cohort. This suggests that QTc is more likely a reflection of cardiac dysfunction and electrolyte abnormalities in a severely ill patient and warrants further study. None of the earlier studies with FCZ doses ranging from 400 to1200mg reported routine QTc monitoring. Pfizer has issued a precaution regarding QTc prolongation in severely ill patients, and concomitant medication known to prolong QTc should be avoided in doses greater than 400mg daily. Routine ECG monitoring was not done in the 2 published studies that used 1200mg FCZ and there was no report of any QTc abnormality, which could have been overlooked [4, 26]. Our study excluded concomitant drugs known to prolong the QTc. We recommend routine QTc monitoring when using high dose FCZ and avoidance of concomitant medications known to prolong the QTc.

Fluconazole is a potent inhibitor of cytochrome P450 [27, 28]. There were no serious hepatic adverse events in any of the FCZ cohorts as was the case in the AMB cohort. Rifampicin was permitted in Stage-2 in participants receiving high dose FCZ and the dose of FCZ was increased in the consolidation phase. Caution and appropriate dose adjustment are still advised when using concomitant drugs that undergo hepatic metabolism and are known to reduce FCZ serum (and corresponding CSF) levels.

The addition of flucytosine to high dose FCZ induction dose may improve the efficacy, but the additional toxicity may mitigate any potential benefit [28–30]. This was studied in the Advancing Cryptococcal Meningitis Treatment for Africa (ACTA) trial, a large study of 721 participants with CM conducted across 9 centres in sub-Saharan Africa in 3 cohorts: flucytosine combined with 1200mg oral FCZ for 2 weeks, AMB combined with 1200mg FCZ or flucytosine for 1 week followed by FCZ for a further week, and AMB combined with flucytosine or 1200mg FCZ for 2 weeks [28]. After 2 weeks, all participants received standard FCZ 400 mg daily. Antiretroviral therapy was started at 4 weeks and participants were followed for a total of 10 weeks. The mortality was lowest in the group receiving AMB combined with flucytosine for

1 week. The French National Agency for Research on AIDS (ANRS) 12257 study conducted in Burundi and Ivory Coast combined 1600mg oral FCZ with flucytosine for 2 weeks followed by 800mg FCZ in the consolidation phase in a single arm study [23]. The preliminary report found mortality of 48'8% at 10 weeks. This is much higher than in our study or the ACTA study [28] and warrants further analysis. Our study enrolled participants with a higher GCS which may partly account for the better outcome and emphasises the need to recognise and treat CM early.

This study is a phase II study and has several limitations. The long duration of the study may account for some of the observations but we could not identify any specific issues. Only participants who could swallow tablets were enrolled implied that potential participants who were extremely ill and with significantly impaired levels of consciousness were not enrolled. We demonstrated a lower mortality compared to other published studies of CM. This may be attributed to the relatively less severe CM illness. It reinforces the view that early recognition and treatment of CM is important. It also shows that oral treatment is feasible in the majority of patients despite the high pill burden with higher doses FCZ.

Whilst flucytosine was permitted this was not available anywhere except in the USA site. This exposes the reality of challenges with optimal treatment access in regions where disease burden is high and reinforces the need for alternative strategies whilst access for optimal treatment is explored. At one point, the study was temporarily halted because of the shortage of study provided FCZ. This highlights the challenges of treating CM in resource limited regions.

The 1200mg FCZ cohort enrolled participants with a lower BMI and lower GCS and this may partially explain the higher mortality. This occurred by chance as enrolment into the increasing dose cohorts was sequential and enrolment was halted after each cohort until safety was assessed by the SMC. This was mandatory to ensure safety of participants before escalating enrolment to the next dose of FCZ. The mortality was still lower than that in published studies of 1200mg FCZ. This does not detract from the observation that 1200mg FZC in the induction phase of CM treatment is inadequate. Fluconazole doses in the induction phase was weight and eGFR adjusted as per protocol. This was considered mandatory for safety because of the higher doses of FZC studied, particularly in respect of QTc prolongation. This is the same protocol for standard of care use of AMB.

## Conclusion

This study demonstrates that weight-based and eGFR adjusted doses of FCZ between 1600mg/day and 2000mg/day are safe, well tolerated and effective in the induction phase of CM treatment and had similar times to achieve CSF sterilization, but took longer than AMB. Survival at these doses of FCZ were not statistically different than AMB. The WHO recommended dose of 1200mg FCZ in the induction phase was associated with poorer survival. The 2000mg FCZ was associated with more gastrointestinal side effects and offered no survival benefit over the 1600mg FCZ dose. A phase III study powered to show non-inferiority of higher doses FCZ to AMB is proposed. We recommend 1600mg FCZ in settings where AMB is not available, based on the current data. Future studies should investigate the addition of flucytosine to the higher FCZ doses and the efficacy of the newer, less toxic antifungal agents. Ultimately prevention and control of HIV which will reduce or eradicate the scourge of CM should be the universal goal.

## Supporting information

**S1 Checklist. CONSORT 2010 checklist of information to include when reporting a randomised trial**\*.
(DOC)

**S1 File.**
(DOC)

## Acknowledgments

We thank the following A5225 researchers and research teams for their contribution to the study: Mohammed Rassool, MBCHB and Noluthando Mwelase, MBCHB- University of the Witwatersrand Helen Joseph Hospital Clinical Research Site (CRS), Dr Rosalina Mnqibisa MBChB, and Dr Mergan Naidoo MBChB, M.Fam.Med, MSc, PhD- Durban International CRS, San Miguel CRS, Alejandro Sanchez, MD & Hannah Edmondson, RN, MPH- University of Southern California CRS, Sandra Rwambuya, MPH and Aggrey Bukuru, MBChB- Joint Clinical Research Centre (JCRC)/Kampala CRS, Kenya Medical Research Institute/Walter Reed Project Clinical Research Center (KEMRI/WRP) CRS, Abraham Mosigisi Siika -MBChB, MMed, MS and David Kiplimo Lagat- MBChB, MMed- Moi University Clinical Research Center (MUCRC) CRS, Byramjee Jeejeebhoy Government Medical College CRS, Patcharaphan Sugandhvesa, M.D. and Daralak Tavornprasit, M.Sc. -Chiang Mai University HIV Treatment, John MacRae, MD, Asociacion Civil Impacta Salud y Educación and Eduardo Ticona, MD Hospital Nacional Dos de Mayo. Lima, Peru

We express a special note of appreciation to all participants and their families and supporters for participation in this study.

## Author Contributions

**Conceptualization:** Umesh G. Lalloo, Judith A. Aberg, David B. Clifford, Robert A. Larsen.

**Data curation:** Lauren Komarow, Evelyn Hogg, Ashley McKhann.

**Formal analysis:** Umesh G. Lalloo, Lauren Komarow, Judith A. Aberg, David B. Clifford, Evelyn Hogg, Ashley McKhann, Robert A. Larsen.

**Funding acquisition:** Umesh G. Lalloo, Evelyn Hogg, Robert A. Larsen.

**Investigation:** Umesh G. Lalloo, Judith A. Aberg, Aggrey Bukuru, David Lagat, Sandy Pillay, Khuanchai Supparatpinyo, Wadzanai Samaneka, Deborah Langat, Eduardo Ticona, Sharlaa Badal-Faesen, Robert A. Larsen.

**Methodology:** Umesh G. Lalloo, Judith A. Aberg, David B. Clifford, Robert A. Larsen.

**Project administration:** Umesh G. Lalloo, Evelyn Hogg, Robert A. Larsen.

**Resources:** Umesh G. Lalloo, Judith A. Aberg, Evelyn Hogg, Robert A. Larsen.

**Supervision:** Umesh G. Lalloo.

**Validation:** Lauren Komarow, Ashley McKhann.

**Visualization:** Umesh G. Lalloo, Judith A. Aberg, Evelyn Hogg, Robert A. Larsen.

**Writing – original draft:** Umesh G. Lalloo.

**Writing – review & editing:** Umesh G. Lalloo, Lauren Komarow, Judith A. Aberg, David B. Clifford, Evelyn Hogg, Ashley McKhann, Aggrey Bukuru, David Lagat, Sandy Pillay, Vidya Mave, Khuanchai Supparatpinyo, Wadzanai Samaneka, Deborah Langat, Eduardo Ticona, Sharlaa Badal-Faesen, Robert A. Larsen.

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
