## [Decision Letter · Decision Letter 0]

26 Nov 2021

PONE-D-21-26982

Higher Dose Oral Fluconazole for the Treatment of AIDS-related Cryptococcal Meningitis (HIFLAC)

PLOS ONE

Dear Dr. Lalloo,

Thank you for submitting your manuscript to PLOS ONE. After careful consideration, we feel that it has merit but does not fully meet PLOS ONE’s publication criteria as it currently stands. Therefore, we invite you to submit a revised version of the manuscript that addresses the points raised during the review process.

We look forward to receiving your revised manuscript.

Kind regards,

Alan Winston

Academic Editor

PLOS ONE

“Research reported in this publication was supported by the National Institute of Allergy and Infectious Diseases of the National Institutes of Health (NIH) under Award Number UM1 AI068634, UM1 AI068636 and UM1 AI106701; clinical trial aidsinfo.nih.gov/clinical-trials/details/NCT00885703. The content is solely the responsibility of the authors and does not necessarily represent the official views of the National Institutes of Health. FCZ was provided by Pfizer.

We thank the following A5225 researchers and research teams for their contribution to the study: Mohammed Rassool, MBCHB and Noluthando Mwelase, MBCHB- University of the Witwatersrand Helen Joseph (WITS HJH) CRS (11101) CTU Grant AI069463, Dr Rosalina Mnqibisa MBChB, and Dr Mergan Naidoo MBChB, M.Fam.Med, MSc, PhD- Durban International CRS (11201) CTU Grant UM1AI069432, 11302, San Miguel CRS CTU Grant AI069438, Alejandro Sanchez, MD & Hannah Edmondson, RN, MPH- University of Southern California CRS (1201) CTU Grant UM1AI069432 and 5 U01 AI069428, Sandra Rwambuya, MPH and Aggrey Bukuru, MBChB- Joint Clinical Research Centre (JCRC)/Kampala CRS (12401) CTU Grant UM1AI069501, 12501, Kenya Medical Research Institute/Walter Reed Project Clinical Research Center (KEMRI/WRP) CRS CTU Grant UM1AI108568, Abraham Mosigisi Siika -MBChB, MMed, MS and David Kiplimo Lagat- MBChB, MMed- Moi University Clinical Research Center (MUCRC) CRS (12601) Grant UM1AI10856, 30313, Parirenyatwa CRS CTU Grant UM1AI069521, 31441, Byramjee Jeejeebhoy Government Medical College CRS CTU Grant UM1AI069465, Patcharaphan Sugandhvesa, M.D. and Daralak Tavornprasit, M.Sc. -Chiang Mai University HIV Treatment (CMU HIV Treatment) CRS (31784) Grant Number: 5UM1AI069399-12, John MacRae, MD,  Asociacion Civil Impacta Salud y Educación and  Eduardo Ticona, MD Hospital Nacional Dos de Mayo. Lima, Peru Grant number AI069438

 We express a special note of appreciation to all participants and their families and supporters for participation in this study.”

“Research reported in this publication was supported by the National Institute of Allergy and Infectious Diseases of the National Institutes of Health (NIH) under Award Number UM1 AI068634, UM1 AI068636 and UM1 AI106701; clinical trial aidsinfo.nih.gov/clinical-trials/details/NCT00885703. The content is solely the responsibility of the authors and does not necessarily represent the official views of the National Institutes of Health. FCZ was provided by Pfizer.”

“No authors have competing interests”

8. We note you have included a table to which you do not refer in the text of your manuscript. Please ensure that you refer to Table 7 in your text; if accepted, production will need this reference to link the reader to the Table.

Reviewers' comments:

Reviewer's Responses to Questions

**Comments to the Author**

1. Is the manuscript technically sound, and do the data support the conclusions?

Reviewer #1: No

Reviewer #2: Yes

Reviewer #3: Yes

2. Has the statistical analysis been performed appropriately and rigorously? 

Reviewer #1: No

Reviewer #2: Yes

Reviewer #3: Yes

3. Have the authors made all data underlying the findings in their manuscript fully available?

Reviewer #1: Yes

Reviewer #2: No

Reviewer #3: Yes

4. Is the manuscript presented in an intelligible fashion and written in standard English?

Reviewer #1: Yes

Reviewer #2: Yes

Reviewer #3: Yes

5. Review Comments to the Author

Reviewer #1: This is a complex but interesting randomised study which questions the recommendation by WHO to use 1200mg/day of fluconazole (FCZ) in the induction phase of cryptococcal meningitis (CM) in person living with HIV (PLWH) in settings where amphotericin-B (AMB) is unavailable. The authors conclude that weight-based induction doses of FCZ between 1600mg/day and 2000mg/day are safe, well tolerated and that the higher dose of 1600mg FCZ should be recommended in settings where AMB is not available on the basis of no statistical difference compared to AMB in mortality. In contrast the WHO recommended 1200mg FCZ dose was associated with poorer survival.

I have a number of concerns:

1. The 1200mg FCZ dosage was ruled out by SMC in Stage 1 for lack of efficacy, so there is no randomised comparison between this dosage and the others.

2. The FCZ 1600mg/day and 2000mg/day dosages were weight-adjusted so effectively a proportion of people actually received lower doses (as low as 1000mg/day). What was the proportion of people receiving <1600mg/day and <2000mg/day in the two arms? I recommend a per-protocol analysis also to be conducted.

3. As often happens when conducting a trial across multiple resource limited international sites there are delays in implementation of the protocol and treatment guidelines are likely to change over the duration of the study creating challenges, such as the addition of the AMB arm in 2011. Also Stage-1 and Stage-2 participants have been combined. One possible consequence could be the observed imbalance between the randomised groups in key confounders such as age and weight and CrAG titer as shown in Table 2. In particular people randomised at 1600mg/day were on average 3 years younger and 4 Kg lighter than those randomised to AMB. CrAG titer was much lower in the FCZ arm. These differences could have biased the comparison. I suggest that the results of a Cox regression after controlling for these imbalances is added in the Results.

4. Statistical power is also an issue. 1600mg/day is judged to be non-inferior to AMB although the difference in mortality at 24 weeks was 6% (30% vs 24%) which does not appear negligible. There is evidence that ABM is superior to FCZ 1600mg/day to achieve a negative culture. This seems to be the most important mediator for survival suggesting that the lack of significance (p>0.14) is only the result of lack of statistical power and cannot be interpreted as ‘non-inferiority’.

5. To increase readability the pairwise comparison p-values of FCZ with AMB (shown in table 5, not Table 3 as indicated on page 9) should be added to the KM figures.

Reviewer #2: This is an interesting study assessing higher dose of fluconazole for naive patients with CM. In several LRCs, given the absence of flucytosine and, often, of AMB the results may have significant clinical implications.

Yet I have some comments:

ABSTRACT and THROUGHOUT the STUDY: patient disposition is quite complex and the lack of data on the 1200 mg arm in the results section of the abstract is somehow catching the eye. I am also confused on the complex study history: while it is needed for explaining the long duration and accrual it is not entirely clear to me.

BACKGROUND:

X Maybe citing liposomial AMB could be beneficial dspite its lack in LRCs

X Several guidelines (including NIH) suggest FCZ 800 mg as consolidation phase: maybe a comment on this can be added.

METHODS:

X No mention of the randomization process is included (block? any sratification?)

X Able to take oral medication is a key inclusion criteria but it could have also been a selection bias (is it available the number of participants not included for such reason?)

X 8 patients received rifampicin (active TB? TBM?) did they differ in terms of outcome?

X there is no mention of IRIS cases and associated mortality

X I coulod not find the statement on data availability as for PlosOne policy

RESULTS AND DISCUSSION:

X The impairment in study participants at baseline is, although by chance, a serious problem for understanding the study outcomes. Specifically patients on FCZ 1200 mg had several features associated with worse outcomes. Is it possible to create a multivariate model including treatment arm and baseline features? However acknowledgeing this in the discussion is, in my opinion, important.

X I was not able to find concomitant oportunistic infections according to study arms (since they may affect mortality and HAART initiation)

X CFU decrease according to study arm: are pairwise comparisons significant? FCZ 1200 and FCZ 2000 seem pretty similar in terms of CFU reduction

X Plasma PK data have not being included: are PK studies ongoing? Will they be available?

X Low eGFR is a risk factor for severe comorbidities and worse outcomes: were participants receiving eGFR-adjusted doses at higher risk of worse outcomes and poor antifungal reponse?

X Discussion pag-12: I believe that acknoweldging the potential impact of several other factors on mortality (as suggested in the last lines of page 14) is crucial for having a fair discussion.

Reviewer #3: This paper with title "Higher Dose Oral Fluconazole for the Treatment of AIDS-related Cryptococcal

Meningitis (HIFLAC)" describes an important piece of research on optimal fluconazole dosing in cryptococcal meningitis.

I congratulate the authors for the work and efforts done to complete this lengthy and laborious trial, which gives important insight on the suboptimal efficacy of what is considered standard of care for Cryptococcal meningitis in resource limited settings (due to unavailability of Ambisome and flucytosine).

I have a number of suggestions /questions to address as minor changes:

The title is concise and appropriate.

1. ABSTRACT - it summarises well the main findings however in the findings part "FINDINGS: 168/154 (safety/efficacy) participants were enrolled with 48/46, 50/45, and 48/43 in the AMB, 1600mg and 2000mg cohorts" I find confusing/unclear how to interpret the results not having read the whole article( eg why more patients were evaluated for safety than efficacy?)

I'd suggest to rephrase or select only to describe efficacy data for instance.

2. In METHODS -

2a The study was approved by the USA Food and Drug Administration and country specific IRBs.-> IRBs needs specification

2b Line "Participants were enrolled in consecutive cohorts at each dose level in Stage-1 and randomized to receive FCZ or AMB in a 3:1 ratio" -> it can be confusing when reading re:consecutive cohorts (as one might think it is the same patient randomised to different doses) - suggest to better clarify what is meant by consecutive cohorts

2c. Line "In Stage-2, participants were concurrently randomized 1:1:1 to receive one of two daily doses (1600mg and 2000mg) of FCZ or AMB". Missing a full stop and would advise substituting “receive one of two daily doses” with “receive either 1600 mg or 2000 mg of fluconazole or AMB” for ease of reading.

2d. Line "In both stages treatment was up to 4 steps followed on study" again unclear, would rephrase

2e. Treatment - Line "Rifampin containing TB treatment was not permitted in the 1200mg FCZ induction cohort. Participants receiving rifampin in the consolidation phase (Step 3 and Step 4) had to take at least 1600mg/day FCZ. There were 4 participants with TB on rifampin in the 1600mg and 2000mg cohort" -> this patients should not be used for the efficacy analysis as fluconazole exposures are significantly reduced by rifampicin co-administration (22% decrease in AUC, 17% decrease in Cmax and 30% higher elimination rate- D panmovana Clin Pharmacokin. 2004 ) - those patients were effectively on lower fluc doses.

2f. Stopping rules - Line "Three or fewer participants with a toxicity event permitted escalation to the next dose; 4-9 instances did not permit escalation, and this would be regarded as the maximum tolerated dose (MTD); more than 9 instances would indicate that the prior dose would be the MTD.” Again I find this difficult to read, unclear. Please rephrase to explain better. What is it mean by instances?

2g. Sample size - the following is difficult to read and unclear, to be clarified/rephrased “This phase 1/2 studywas designed to have 90% power and a one-sided 0.10 alpha test to detect a difference in mean change in log10 CFU/mL CSF/day of about 80% of the standard deviation of the measures in stage-1A sample size of 24 participants in the FCZ arm of each Stage-1 cohort would detect a dose limiting toxicity (DLT) rate > 25% of participants and would provide guidance on efficacy for the choice of cohorts in Stage-2.”

2h. the line “Additionally, with >40 evaluable participants per dose, there is an 87% chance of seeing one or more rare events if the unacceptable event occurs at a rate of 5% or higher”please clarify what is meant by unacceptable event

3. RESULTS

3a I personally found the results section difficult to read, mostly describing what is found/seen in the relative tables and figures however the main findings should be described clearly in the text with reference to tables/figures for the complete findings. I would suggest to work on the result section to improve the description of the findings.

3b Line “Several participants with hyponatremia and hyperkalaemia received a glucocorticosteroid: 9/25 FCZ 1600 mg, 8/23 FCZ 2000mg, and 9/19 AMB" . It would be important to specify what glucocorticoid was given and at what dose, as this could potentially be confounding the results.

FIGURES

Figure 1. following sentence needs checking “Participants from Stage 1 for each of the 1600mg and 2000mg FCZ and AMB cohorts were combined with the participants”

Figure 2. correct the sentence “There was no difference when participants who failed fluconazole and were switched to amphotericin B”

6. PLOS authors have the option to publish the peer review history of their article (what does this mean?). If published, this will include your full peer review and any attached files.

Reviewer #1: No

Reviewer #2: No

Reviewer #3: **Yes: **Margherita Bracchi

---

## [Author Response · Author response to Decision Letter 0]

12 Jul 2022

Alan Winston

Academic Editor

PLOS ONE

11th April 2022

Dear Sir

RESUBMISSION: Higher Dose Oral Fluconazole for the Treatment of AIDS-related Cryptococcal Meningitis (HIFLAC) – PONE-D-21-26982

Kindly note that I have made all revisions recommended following review of the manuscript PONE-D-21-26982. I trust it will have the approval of the reviewers and Editorial Board of PLOS-ONE. A separate letter labelled “Response to Reviewers” has been uploaded.

The authors have declared that no competing interests exist. 

Research reported in this publication was supported by the National Institute of Allergy and Infectious Diseases of the National Institutes of Health under Award Number UM1 AI068634, UM1 AI068636 and UM1 AI106701; clinical trial aidsinfo.nih.gov/clinical-trials/details/NCT00885703. The content is solely the responsibility of the authors and does not necessarily represent the official views of the National Institutes of Health. FCZ was provided by Pfizer. 

Repository information in respect of this study will be provided at time of acceptance of the manuscript.

I look forward to acceptance of the manuscript.

Umesh Lalloo

RESPONSE TO REVIEWERS: Higher Dose Oral Fluconazole for the Treatment of AIDS-related Cryptococcal Meningitis (HIFLAC) – PONE-D-21-26982

Thank you for the critical review of above manuscript.

The responses to the reviewers is itemised and I have attached a copy of the original manuscript with the changes tracked and a final unmarked version.

A. ACADEMIC EDITOR’S REVIEW

1. The manuscript has been revised to comply with PLOS ONE formatting style and title authors affiliations.

2. Tables have been included as part of the main manuscript

3. The Funding information has been corrected and in the Funding Information section.

4. Funding information has been removed from the “Acknowledgements” section of the manuscript.

5. The online submission form now states “No authors have competing interests”.

6. Repository information has been provided in the form of a PDF document titled: A5225 primary analysis.

7. A full ethics statement has been made in the “Methods” section. The IRB committees that approved the protocol are from the institutions that have been named in this section.

8. The Table 7 that has not been referred to in the text has been corrected in the revised manuscript.

B. REVIEWERS COMMENTS:

Reviewer #1: This is a complex but interesting randomised study which questions the recommendation by WHO to use 1200mg/day of fluconazole (FCZ) in the induction phase of cryptococcal meningitis (CM) in person living with HIV (PLWH) in settings where amphotericin-B (AMB) is unavailable. The authors conclude that weight-based induction doses of FCZ between 1600mg/day and 2000mg/day are safe, well tolerated and that the higher dose of 1600mg FCZ should be recommended in settings where AMB is not available on the basis of no statistical difference compared to AMB in mortality. In contrast the WHO recommended 1200mg FCZ dose was associated with poorer survival.

I have a number of concerns:

1. The 1200mg FCZ dosage was ruled out by SMC in Stage 1 for lack of efficacy, so there is no randomised comparison between this dosage and the others.

It was not the intention per protocol to undertake a randomised comparison between the weight-based 1200mg FCZ cohort and the two cohorts, 1600mg and 2000mg that were deemed safe in Stage 1. The 1200mg cohort was stopped when the mortality threshold was reached in terms of the stopping rules developed for Stage 1. A mortality of 41% occurred in this first cohort after the enrolment of 22 participants which was 2 less than the predetermined sample size for the Stage 1 cohorts. I have for reference therefore added this cohort in the Kaplan Meier plots for time to mortality and CFS sterilization

2. The FCZ 1600mg/day and 2000mg/day dosages were weight-adjusted so effectively a proportion of people actually received lower doses (as low as 1000mg/day). What was the proportion of people receiving <1600mg/day and <2000mg/day in the two arms? I recommend a per-protocol analysis also to be conducted. 

It is correct that many participants did receive lower than the 1200mg, 1600mg and 2000mg doses of FCZ in the respective groups as this was determined per protocol as part of the safety of FCZ administration at higher doses. It would be difficult to interpret an analysis based on actual doses received. In the event higher doses of FCZ are recommended as standard of care it should always be weight based and adjusted for GFR for safety reasons. It must also be noted that AMB is also administered weight-based in standard of care protocols and none of the studies published actually analyse the data by actual dose. The actual dose received represents a biologically correct dose based on the area under the curve of a pharmacokinetic analysis.

3. As often happens when conducting a trial across multiple resource limited international sites there are delays in implementation of the protocol and treatment guidelines are likely to change over the duration of the study creating challenges, such as the addition of the AMB arm in 2011. Also Stage-1 and Stage-2 participants have been combined. One possible consequence could be the observed imbalance between the randomised groups in key confounders such as age and weight and CrAG titer as shown in Table 2. In particular people randomised at 1600mg/day were on average 3 years younger and 4 Kg lighter than those randomised to AMB. CrAG titer was much lower in the FCZ arm. These differences could have biased the comparison. I suggest that the results of a Cox regression after controlling for these imbalances is added in the Results.

As noted by the reviewer, we concede that imbalances in the groups that are relatively smaller than those for phase 3 clinical trials are likely to occur. We believe that apart from the age difference the study has controlled for some of these variables by adjusting for weight and GFR. Participants were not stratified according to disease severity because a) in Stage 1 the participants were enrolled sequentially and after the preceding dose cohort was completed or as in the case of the 1200mg cohort, prematurely halted, and b) in Stage 2 participants were randomised into the 3 cohorts that were enrolled simultaneously. As indicated this was a phase 2 dose ranging safety and efficacy study to determine optimal doses for a larger phase 3 study.

4. Statistical power is also an issue. 1600mg/day is judged to be non-inferior to AMB although the difference in mortality at 24 weeks was 6% (30% vs 24%) which does not appear negligible. There is evidence that ABM is superior to FCZ 1600mg/day to achieve a negative culture. This seems to be the most important mediator for survival suggesting that the lack of significance (p>0.14) is only the result of lack of statistical power and cannot be interpreted as ‘non-inferiority’.

We agree that AMB appears to be superior to the 1600mg FCZ dose to achieve a negative CSF culture, but based on the sample size one cannot state that it is indeed superior. We believe that the data from this study provides justification for a large phase 3 study using 1600mg FCZ, 2000mg FCZ and AMB. I think that we will have equipoise in deciding whether to use 1600mg or 2000mg FCZ in a phase 3 trial.

5. To increase readability the pairwise comparison p-values of FCZ with AMB (shown in table 5, not Table 3 as indicated on page 9) should be added to the KM figures.

This has been done.

Reviewer #2: This is an interesting study assessing higher dose of fluconazole for naive patients with CM. In several LRCs, given the absence of flucytosine and, often, of AMB the results may have significant clinical implications.

Yet I have some comments:

ABSTRACT and THROUGHOUT the STUDY: patient disposition is quite complex and the lack of data on the 1200 mg arm in the results section of the abstract is somehow catching the eye. I am also confused on the complex study history: while it is needed for explaining the long duration and accrual it is not entirely clear to me.

This comment is similar to the point raised by Reviewer 1 and I have included the 1200mg FCZ cohort data in the Kaplan Meier plot figures. It must be emphasised that per protocol the 1200mg FCZ cohort was considered ineffective due to a higher mortality and therefore not included in Stage 2. Further, the number of participants in the 1200mg arm was only 22 and 50 and 48 in the pooled date for the 1600mg and 2000mg FCZ cohorts, tuis making comparisons difficult to interpret..

BACKGROUND:

X Maybe citing liposomial AMB could be beneficial dspite its lack in LRCs.

We agree this is an important point but also acknowledge the high cost of liposomal AMB in resource limited settings where there is restricted access to standard formulation of AMB. This has been addressed in the Introduction in the revised manuscript.

X Several guidelines (including NIH) suggest FCZ 800 mg as consolidation phase: maybe a comment on this can be added.

This has also been addressed in the Introduction section in the revised manuscript.

METHODS:

X No mention of the randomization process is included (block? any sratification?)

This has been in clarified in the Methods section. Participants were enrolled in Stage 1 in the sequential cohorts and block randomised. In Stage 2 they were block randomised to any of the 3 arms viz 1600mg and 2000mg FCZ and AMB.

X Able to take oral medication is a key inclusion criteria but it could have also been a selection bias (is it available the number of participants not included for such reason?)

We did collect this data. Based on the database no participants were screened who could not take oral medication.

X 8 patients received rifampicin (active TB? TBM?) did they differ in terms of outcome?

X there is no mention of IRIS cases and associated mortality

Thank you for raising this point as it should be presented in the manuscript. Two participants developed IRIS, one in the 1200mg arm who commenced ART on day 63 and developed IRIS on day 99; and one in the AMB arm commenced ART on day 26 and developed IRIS on day 38. Both survived. This has been included in the Results and referred to in the Discussion.

X I coulod not find the statement on data availability as for PlosOne policy.

This has been done.

RESULTS AND DISCUSSION:

X The impairment in study participants at baseline is, although by chance, a serious problem for understanding the study outcomes. Specifically patients on FCZ 1200 mg had several features associated with worse outcomes. Is it possible to create a multivariate model including treatment arm and baseline features? However acknowledgeing this in the discussion is, in my opinion, important.

This has been acknowledged in the Discussion.

X I was not able to find concomitant oportunistic infections according to study arms (since they may affect mortality and HAART initiation)

This is acknowledged. There was no systematic trends in any of the OIs and none were clinically severe as this would have been exclusionary. The OIs and ADIS defining conditions present at enrolment are presented in the Table 3 in the manuscript. Whilst there frequency was higher in the 1200mg cohort this was not significant.

 X CFU decrease according to study arm: are pairwise comparisons significant? FCZ 1200 and FCZ 2000 seem pretty similar in terms of CFU reduction.

This is correct.

X Plasma PK data have not being included: are PK studies ongoing? Will they be available?

Plasma PK values have not been done and may be analysed in the future.

X Low eGFR is a risk factor for severe comorbidities and worse outcomes: were participants receiving eGFR-adjusted doses at higher risk of worse outcomes and poor antifungal reponse?

There was no significant difference in outcomes in participants with reduced eGFR.

X Discussion pag-12: I believe that acknoweldging the potential impact of several other factors on mortality (as suggested in the last lines of page 14) is crucial for having a fair discussion.

The impact of other factors such as eGFR, GCS, BMI, opportunistic infections, and CD4 count on outcomes have been added to the Discussion.

Reviewer #3: This paper with title "Higher Dose Oral Fluconazole for the Treatment of AIDS-related Cryptococcal

Meningitis (HIFLAC)" describes an important piece of research on optimal fluconazole dosing in cryptococcal meningitis.

I congratulate the authors for the work and efforts done to complete this lengthy and laborious trial, which gives important insight on the suboptimal efficacy of what is considered standard of care for Cryptococcal meningitis in resource limited settings (due to unavailability of Ambisome and flucytosine).

I have a number of suggestions /questions to address as minor changes:

The title is concise and appropriate.

1. ABSTRACT - it summarises well the main findings however in the findings part "FINDINGS: 168/154 (safety/efficacy) participants were enrolled with 48/46, 50/45, and 48/43 in the AMB, 1600mg and 2000mg cohorts" I find confusing/unclear how to interpret the results not having read the whole article( eg why more patients were evaluated for safety than efficacy?)

I'd suggest to rephrase or select only to describe efficacy data for instance.

This has been changed as recommended in the abstract. It has been reworded to report 168 enrolled participants.

2. In METHODS -

2a The study was approved by the USA Food and Drug Administration and country specific IRBs.-> IRBs needs specification

The individual IRBs have been presented in the Methods section as per PLOS ONE requirements.

2b Line "Participants were enrolled in consecutive cohorts at each dose level in Stage-1 and randomized to receive FCZ or AMB in a 3:1 ratio" -> it can be confusing when reading re:consecutive cohorts (as one might think it is the same patient randomised to different doses) - suggest to better clarify what is meant by consecutive cohorts

This has been done.

2c. Line "In Stage-2, participants were concurrently randomized 1:1:1 to receive one of two daily doses (1600mg and 2000mg) of FCZ or AMB". Missing a full stop and would advise substituting “receive one of two daily doses” with “receive either 1600 mg or 2000 mg of fluconazole or AMB” for ease of reading.

This has been reworded as advised and reads with better clarity.

2d. Line "In both stages treatment was up to 4 steps followed on study" again unclear, would rephrase

It has been rephrased – “Treatment was planned in 4 steps as follows: and “Participants were followed on study for 24 weeks.”

2e. Treatment - Line "Rifampin containing TB treatment was not permitted in the 1200mg FCZ induction cohort. Participants receiving rifampin in the consolidation phase (Step 3 and Step 4) had to take at least 1600mg/day FCZ. There were 4 participants with TB on rifampin in the 1600mg and 2000mg cohort" -> this patients should not be used for the efficacy analysis as fluconazole exposures are significantly reduced by rifampicin co-administration (22% decrease in AUC, 17% decrease in Cmax and 30% higher elimination rate- D panmovana Clin Pharmacokin. 2004 ) - those patients were effectively on lower fluc doses.

We acknowledge this point. We believe that these participants should be included as this was planned per protocol. Its effect will be to dilute the efficacy and therefore strengthen the findings that the 1600mg and 2000mg FCZ dose is more effective. Excluding these participants made no significant difference. These higher doses have not been used before and there is no guideline currently to adjust FCZ doses with concomitant rifampin. If a phase 3 study is planned this would be a consideration. 

This has been presented in the Results section and in the Discussion.

Results.

“Rifampicin was permitted in Stage 2 to facilitate recruitment in high TB incident regions. Rifampicin was commenced in 4 participants in the induction phase of treatment in the FCZ cohort: 1 in the 1600mg cohort commenced on Day 11 and survived. There were 3 in the 2000mg cohort: 1 commenced on day 2, and died on day 80; 1 commenced on day 3 and survived; and 1 commenced on day 10 and died on day 80. No dose adjustments were planned for participants on rifampicin. Excluding these participants from the analysis made no difference to the statistical outcome.

Discussion:

Rifampicin may reduce the bioavailability of FCZ and was only permitted in Stage 2 of the protocol in the 1600mg and 2000mg FCZ cohorts. A pharmacokinetic study of 400mg FCZ in the consolidation phase of CM following 2 weeks of AMB, showed a 39% increase in the elimination constant of serum FCZ with concomtitant rifamipicin at 600mg daily (19). This did not alter the outcomes for CM. Our study did not adjust FCZ doses of the doses and excluding participants on rifampicin did not make any significant didference. We propose that a larger phase 3 study of higher doses of FCZ should take this into account. 

2f. Stopping rules - Line "Three or fewer participants with a toxicity event permitted escalation to the next dose; 4-9 instances did not permit escalation, and this would be regarded as the maximum tolerated dose (MTD); more than 9 instances would indicate that the prior dose would be the MTD.” Again I find this difficult to read, unclear. Please rephrase to explain better. What is it mean by instances?

This has been rephrased and now reads with greater clarity.

2g. Sample size - the following is difficult to read and unclear, to be clarified/rephrased “This phase 1/2 studywas designed to have 90% power and a one-sided 0.10 alpha test to detect a difference in mean change in log10 CFU/mL CSF/day of about 80% of the standard deviation of the measures in stage-1A sample size of 24 participants in the FCZ arm of each Stage-1 cohort would detect a dose limiting toxicity (DLT) rate > 25% of participants and would provide guidance on efficacy for the choice of cohorts in Stage-2.”

We agree this sentence is unclear and clarified in the Methods as follows:

The sample size was designed to have a 90% power and a one-sided 0.10 alpha test to detect a change in log10 CFU//ml CSF of about 80% of the standard deviation of the values in Stage 1. Based on this a sample size of 24 was determined and would also permit a dose limiting toxicity rate of >25%. This permitted the identification of the maximum tolerated dose (MTD). The same rationale was used to determine the sample size in Stage 2 once the MTD was identified. In the conduct of Stage 1, the 1200mg FCZ dose was associated with a high mortality and excluded from Stage 2. Up to the maximum dose of 2000mg FCZ there was no MTD, hence the 1600mg and 2000mg FCZ doses were tested in Stage 2. 

2h. the line “Additionally, with >40 evaluable participants per dose, there is an 87% chance of seeing one or more rare events if the unacceptable event occurs at a rate of 5% or higher”please clarify what is meant by unacceptable event.

Unacceptable event refers to a Grade 3 or 4 DAIDS toxicity. This has been clarified in the Methods section.

3. RESULTS

3a I personally found the results section difficult to read, mostly describing what is found/seen in the relative tables and figures however the main findings should be described clearly in the text with reference to tables/figures for the complete findings. I would suggest to work on the result section to improve the description of the findings.

This has been done.

3b Line “Several participants with hyponatremia and hyperkalaemia received a glucocorticosteroid: 9/25 FCZ 1600 mg, 8/23 FCZ 2000mg, and 9/19 AMB" . It would be important to specify what glucocorticoid was given and at what dose, as this could potentially be confounding the results.

The glucocorticoid given was hydrocortisone, prednisone, cortisone actetate and betamethasone, and given for presumed adrenal toxicity from the higher doses of FCZ. However this was not significantly different from the AMB cohort. We therefore do not think this would have confounded the results.

FIGURES

Figure 1. following sentence needs checking “Participants from Stage 1 for each of the 1600mg and 2000mg FCZ and AMB cohorts were combined with the participants”

This has been corrected. 

“Participants from Stage 2 for each of the 1600mg and 2000mg FCZ and AMB cohorts were combined with the participants from Stage 1.”

Figure 2. correct the sentence “There was no difference when participants who failed fluconazole and were switched to amphotericin B”

This has been corrected as follows:

In the 1600mg FZ arm, 4/45 subjects switched to Ampho B and culture converted by week 10, An additional 5/45 started ampho B and subsequently died.

In the 2000mg FZ arm 5/43 subjects switched to Ampho B and culture converted by week 10. An additional 3/43 started ampho B and subsequently died.

In the Ampho B arm, 1/46 subjects switched to Fluconazole and culture converted by week 10.

I trust that we have responded constructively to the reviewers and believe that the manuscript is now acceptable for publication in PLOS ONE.

Yours sincerely

U G Lalloo – Corresponding author.

---

## [Decision Letter · Decision Letter 1]

18 Oct 2022

PONE-D-21-26982R1Higher Dose Oral Fluconazole for the Treatment of AIDS-related Cryptococcal Meningitis (HIFLAC)PLOS ONE

Dear Dr. Lalloo,

Thank you for submitting your manuscript to PLOS ONE. After careful consideration, we feel that it has merit but does not fully meet PLOS ONE’s publication criteria as it currently stands. Therefore, we invite you to submit a revised version of the manuscript that addresses the points raised during the review process. Two of the previous reviewer were available to provide further comments on your study and were happy with your revision. However it was considered necessary to obtain comments from an additional reviewer. The reviewer's comments are available below. The reviewer raised some scientific concerns about the study that need to be addressed in a revision.

Please revise the manuscript to address all the reviewer's comments in a point-by-point response in order to ensure it is meeting the journal's publication criteria. Please note that the revised manuscript will need to undergo further review, we thus cannot at this point anticipate the outcome of the evaluation process.

We look forward to receiving your revised manuscript.

Kind regards,

Miquel Vall-llosera Camps

SeniorEditor

PLOS ONE

Journal Requirements:

Reviewers' comments:

Reviewer's Responses to Questions

**Comments to the Author**

1. If the authors have adequately addressed your comments raised in a previous round of review and you feel that this manuscript is now acceptable for publication, you may indicate that here to bypass the “Comments to the Author” section, enter your conflict of interest statement in the “Confidential to Editor” section, and submit your "Accept" recommendation.

Reviewer #2: All comments have been addressed

Reviewer #3: All comments have been addressed

Reviewer #4: All comments have been addressed

2. Is the manuscript technically sound, and do the data support the conclusions?

Reviewer #2: Yes

Reviewer #3: Yes

Reviewer #4: Partly

3. Has the statistical analysis been performed appropriately and rigorously? 

Reviewer #2: I Don't Know

Reviewer #3: I Don't Know

Reviewer #4: Yes

4. Have the authors made all data underlying the findings in their manuscript fully available?

Reviewer #2: Yes

Reviewer #3: Yes

Reviewer #4: No

5. Is the manuscript presented in an intelligible fashion and written in standard English?

Reviewer #2: Yes

Reviewer #3: Yes

Reviewer #4: Yes

6. Review Comments to the Author

Reviewer #2: Thanks for addressing all my comments. I believe the manuscript is more clear now and all the data have been explained smoothly.

Reviewer #3: Thank you for the revision work performed on the manuscript. All the comments raised and suggestions have been addressed.

Reviewer #4: This study involves a complex study design aiming to determine the maximum tolerated dose and the safety/efficacy of an induction-consolidation strategy of higher doses FCZ. Given the less availability of amphotericin-B (AMB) and flucytosine, the results can be impactful.

1) It is claimed the induction phase weight and renal-adjusted doses of 1600mg and 2000mg/day FCZ for CM were safe and well tolerated according to the statistical testing criteria defined; however, according to the results, for example, Figures 2,3, & 4, the AMB group seem to be systemically superior, even though the statistical test significance levels didn’t reach the defined criteria. One can argue it’s a power / sample size issue. It seems to me the AMB treatment can be regarded as the gold standard, and an inferior test / equivalence test should have been used.

2) There are typos in the manuscript. For example, in Abstract, “proportin”; in the first paragraph of Introduction, “maybe” should be “may be”.

7. PLOS authors have the option to publish the peer review history of their article (what does this mean?). If published, this will include your full peer review and any attached files.

Reviewer #2: No

Reviewer #3: **Yes: **margherita bracchi

Reviewer #4: No

---

## [Author Response · Author response to Decision Letter 1]

14 Dec 2022

REVIEWER #4:

The typos in the manuscript are regretted and have been corrected viz: “proportion” is spelt correctly in the revised manuscript; and “maybe” is now written “may be” in the revised manuscript.

1) It is claimed the induction phase weight and renal-adjusted doses of 1600mg and 2000mg/day FCZ for CM were safe and well tolerated according to the statistical testing criteria defined; however, according to the results, for example, Figures 2,3, & 4, the AMB group seem to be systemically superior, even though the statistical test significance levels didn’t reach the defined criteria. One can argue it’s a power / sample size issue. It seems to me the AMB treatment can be regarded as the gold standard, and an inferior test / equivalence test should have been used.

The study team has considered this point carefully and agree that an inferior/equivalence test may have been useful to guide CM treatment with AMB of FCZ where AMB is not available. This study was a phase I/II study to explore the maximum tolerated dose of FCZ and efficacy with a view to informing doses in a phase III study, where this will certainly be a consideration in the study design. The number of participants in the study is too small for non-inferiority/equivalence study, and equivalence bounds needed to be set a priori. The confidence intervals would also need to inside the non-inferiority bound. 

I have added the following point in the discussion:

“Amphoterecin B appears to be superior to higher doses of FCZ but did not reach statistical significance. A phase III study should be powered to explore non-inferiority of higher doses FCZ to AMB.”

---

## [Decision Letter · Decision Letter 2]

27 Jan 2023

Higher Dose Oral Fluconazole for the Treatment of AIDS-related Cryptococcal Meningitis (HIFLAC)

PONE-D-21-26982R2

Dear Dr. Lalloo,

We’re pleased to inform you that your manuscript has been judged scientifically suitable for publication and will be formally accepted for publication once it meets all outstanding technical requirements.

Kind regards,

Renee Ridzon

Academic Editor

PLOS ONE

Additional Editor Comments (optional):

Reviewers' comments:

Reviewer's Responses to Questions

**Comments to the Author**

1. If the authors have adequately addressed your comments raised in a previous round of review and you feel that this manuscript is now acceptable for publication, you may indicate that here to bypass the “Comments to the Author” section, enter your conflict of interest statement in the “Confidential to Editor” section, and submit your "Accept" recommendation.

Reviewer #2: All comments have been addressed

Reviewer #3: All comments have been addressed

Reviewer #4: All comments have been addressed

2. Is the manuscript technically sound, and do the data support the conclusions?

Reviewer #2: Yes

Reviewer #3: Yes

Reviewer #4: Yes

3. Has the statistical analysis been performed appropriately and rigorously? 

Reviewer #2: Yes

Reviewer #3: Yes

Reviewer #4: Yes

4. Have the authors made all data underlying the findings in their manuscript fully available?

Reviewer #2: Yes

Reviewer #3: Yes

Reviewer #4: No

5. Is the manuscript presented in an intelligible fashion and written in standard English?

Reviewer #2: Yes

Reviewer #3: Yes

Reviewer #4: Yes

6. Review Comments to the Author

Reviewer #2: Thanks for addressing all my comments. I have no other change to require. From my point of view it can be accepted as it is.

Reviewer #3: All queries have been successfully answered in my opinion. I had no additional comments and previously stated that the paper was to be accepted for publication.

Reviewer #4: My comments were addressed. There is no further critique.

My comments were addressed. There is no further critique.

7. PLOS authors have the option to publish the peer review history of their article (what does this mean?). If published, this will include your full peer review and any attached files.

Reviewer #2: No

Reviewer #3: No

Reviewer #4: No

---

## [Editor Report · Acceptance letter]

3 Feb 2023

PONE-D-21-26982R2 

Higher Dose Oral Fluconazole for the Treatment of AIDS-related Cryptococcal Meningitis (HIFLAC) – report of A5225, a multicentre... 

Dear Dr. Lalloo:

I'm pleased to inform you that your manuscript has been deemed suitable for publication in PLOS ONE. Congratulations! Your manuscript is now with our production department. 

Kind regards, 

on behalf of

Dr. Renee Ridzon 

Academic Editor

PLOS ONE